# Targeting lysyl oxidase (LOX) overcomes chemotherapy resistance in triple negative breast cancer

Ozge Saatci[1], Aysegul Kaymak[1], Umar Raza[2], Pelin G. Ersan[2], Ozge Akbulut[2], Carolyn E. Banister[1], Vitali Sikirzhytski[1], Unal Metin Tokat[2], Gamze Aykut[2], Suhail A. Ansari[2], Hayriye Tatli Dogan[3], Mehmet Dogan[4], Pouria Jandaghi[5,6], Aynur Isik[7], Fatma Gundogdu[8], Kemal Kosemehmetoglu[8], Omer Dizdar[9], Sercan Aksoy[9], Aytekin Akyol[7,8], Aysegul Uner[8,10], Phillip J. Buckhaults[1], Yasser Riazalhosseini[5,6] & Ozgur Sahin[1,2✉]

Chemoresistance is a major obstacle in triple negative breast cancer (TNBC), the most aggressive breast cancer subtype. Here we identify hypoxia-induced ECM re-modeler, lysyl oxidase (LOX) as a key inducer of chemoresistance by developing chemoresistant TNBC tumors in vivo and characterizing their transcriptomes by RNA-sequencing. Inhibiting LOX reduces collagen cross-linking and fibronectin assembly, increases drug penetration, and downregulates ITGA5/FN1 expression, resulting in inhibition of FAK/Src signaling, induction of apoptosis and re-sensitization to chemotherapy. Similarly, inhibiting FAK/Src results in chemosensitization. These effects are observed in 3D-cultured cell lines, tumor organoids, chemoresistant xenografts, syngeneic tumors and PDX models. Re-expressing the hypoxia-repressed miR-142-3p, which targets *HIF1A*, *LOX* and *ITGA5*, causes further suppression of the HIF-1α/LOX/ITGA5/FN1 axis. Notably, higher LOX, ITGA5, or FN1, or lower miR-142-3p levels are associated with shorter survival in chemotherapy-treated TNBC patients. These results provide strong pre-clinical rationale for developing and testing LOX inhibitors to overcome chemoresistance in TNBC patients.

[1] Department of Drug Discovery and Biomedical Sciences, University of South Carolina, Columbia, SC 29208, USA. [2] Faculty of Science, Department of Molecular Biology and Genetics, Bilkent University, 06800 Ankara, Turkey. [3] Department of Medical Pathology, Ankara Yildirim Beyazit University, 06800 Ankara, Turkey. [4] Department of Medical Pathology, Ankara Oncology Education and Research Hospital, 06200 Ankara, Turkey. [5] Department of Human Genetics, McGill University, Montreal, QC H3A 1B1, Canada. [6] McGill University and Genome Quebec Innovation Centre, Montreal, QC H3A 0G1, Canada. [7] Hacettepe University Transgenic Animal Technologies Research and Application Center, 06100 Ankara, Turkey. [8] Faculty of Medicine, Department of Pathology, Hacettepe University, 06100 Ankara, Turkey. [9] Department of Medical Oncology, Hacettepe University Cancer Institute, 06100 Ankara, Turkey. [10] Hacettepe University Molecular Pathology Research and Application Center, 06100 Ankara, Turkey. ✉email: sahinozgur@gmail.com

Triple negative breast cancer (TNBC) is the most aggressive subtype of breast cancer. It accounts for 10–20% of all patients, yet is responsible for 30% of all breast cancer deaths[1]. At the molecular level, TNBC is characterized by the lack of estrogen receptor alpha (ERα), progesterone receptor (PR) and epidermal growth factor receptor 2 (HER2/ErbB2)[2,3] expression. TNBC patients mostly rely on chemotherapy unlike other sub-types that can be treated with targeted therapies. Anthracyclines and taxane-based chemotherapy agents are among the most commonly used chemotherapeutics in both neo-adjuvant and adjuvant settings[4]. As compared to other subtypes, TNBC patients show low risk of recurrence if pathological complete response (pCR) is achieved[5]. However, while only 30–40% of TNBC patients show pCR towards treatment, the majority have <60% 5-year survival due to aggressive relapse[6]. Recently, the first immunotherapy for breast cancer was approved by the Food and Drug Administration (FDA) to treat locally advanced or meta-static TNBC patients. However, immunotherapy combined with chemotherapy produced only a 3 month improvement in progression-free survival, and improvement in overall survival was observed only in patients with PD-L1 positive tumors[7]. Therefore, there is still a dire need to identify novel molecular targets to improve the therapeutic benefit of chemotherapy given in first-line settings, as well as in patients with advanced, che-motherapy resistant TNBC. This will have immediate transla-tional impact on improving the pCR rate to standard chemotherapy and will improve patient outcome for the most-aggressive breast cancer subtype.

Deregulation of distinct cell intrinsic processes, such as apop-tosis[8], growth factor signaling[9], DNA repair[10] as well as altera-tions in the levels of drug transporter proteins[11] have previously been associated with chemoresistance. In addition, accumulating evidence suggest that the tumor extracellular matrix (ECM) may also confer resistance to therapy, either by providing a protective barrier that hinders access of anti-cancer drugs to tumors[12–14] or by activating survival signaling or blocking apoptosis upon interacting with the integrin type transmembrane receptors[15,16]. Integrins are heterodimeric cell surface receptors that control cell adhesion, cytoskeletal organization, signal transduction and cell migration via establishing focal adhesion complexes that serve as mechanical links that convey signals between ECM and the intracellular compartment of the interacting cell[17,18]. Activation of integrin signaling may lead to resistance to therapy[15,19,20] and acquisition of metastatic traits[21]. Notably, increased expression of several integrin subunits have previously been associated with poor patient outcome[19,22,23].

The ECM is a highly dynamic structure that is constantly remodeled by cells through altered synthesis, degradation, reas-sembly and chemical modifications[24]. While normal epithelial cells produce small amounts of ECM, fibroblasts and tumors cells produce large quantities of ECM molecules, such as collagen and fibronectin[25,26]. In addition, overexpression of ECM remodeling enzymes such as matrix metalloproteases (ECM degraders) and lysyl oxidases (ECM stiffeners) have previously been associated with tumor aggressiveness[24]. The lysyl oxidase (LOX) family proteins mediate the conversion of lysine residues in collagen and elastin precursors into highly reactive aldehydes thereby trigger-ing cross-linking and stabilization of ECM proteins, specifically the type I collagen and elastin, and regulate cell adhesion, motility and invasion[27]. In preclinical breast cancer models, LOX secre-tion from breast cancer cells was shown to induce the pre-metastatic niche formation[28], and LOX inhibition suppressed lung metastasis[29]. LOX inhibition was also shown to sensitize pancreatic cancer to chemotherapy[30,31]. Furthermore, novel intracellular functions of the LOX family proteins have been reported, such as the stabilization of transcription factors and maintenance of chromosome stability[32]. Despite the relatively well-established roles of LOX in stimulating cancer metastasis, its contribution to chemoresistance in TNBCs has not yet been reported.

In this study, we combine in vivo-developed chemoresistant xenograft models with an unbiased, genome-wide transcriptomic approach, 3D cell culture systems, primary breast cancer orga-noids, syngeneic and patient-derived xenograft (PDX) models and patient data/tissue analyses to identify mechanisms of che-motherapy resistance in TNBCs. We show that chemoresistance is driven by the HIF-1α/miR-142-3p/LOX/ITGA5/FN1 axis and targeting either LOX or its downstream FAK/Src signaling pathway, or overexpressing miR-142-3p can overcome che-moresistance in TNBC.

## Results

**Integrin signaling is a key mediator of chemoresistance in TNBCs.** To elucidate the underlying mechanisms of che-moresistance in TNBCs, we modelled the clinical acquired resistance by using xenografts of the well-established TNBC cell line, MDA-MB-231[33,34]. Tumor-bearing mice were continuously treated with either vehicle or doxorubicin, and the fast-growing vehicle-treated mice were sacrificed, and tumors were denoted as vehicle. When tumors from the doxorubicin-treated group exhibited initial response to therapy and shrunk, tumors from some of the mice were collected and denoted as sensitive. The rest of the mice were kept under doxorubicin treatment until their tumors exhibited re-growth at rates comparable to vehicle-treated tumors, and those tumors were classified as resistant (Fig. 1a). The average growth curves and the Waterfall plot showing tumor volume fold change over time for vehicle-treated, doxorubicin-sensitive and -resistant tumors are depicted in Fig. 1b, c, respectively.

We performed RNA-sequencing (RNA-seq) on bulk tumors (four tumors from each group) to identify transcripts differen-tially expressed between doxorubicin-sensitive and -resistant tumors. We used the 441 most differentially expressed genes (fold change (FC) cut-off = 1.75, $p$-value < 0.05, Supplementary Data 1) to create a doxorubicin resistance gene signature (DoxoR-GS). We then tested its clinical relevance using gene expression profiling datasets of chemotherapy-treated TNBC patients (for details, see "Methods" section)[35]. Chemotherapy-treated TNBC patients from GSE58812[36] that have high DoxoR-GS scores exhibit poor overall survival (OS) as compared to those with low DoxoR-GS scores (Fig. 1d). These results were further confirmed using METABRIC dataset[37] (Supplementary Fig. 1a) and in another dataset (GSE31519) where we showed that expression of DoxoR-GS is significantly higher in patients who exhibit an event, described as either relapse- or distant metastasis (Supplementary Fig. 1b).

Having validated the clinical relevance of our in vivo-derived DoxoR-GS, we sought to identify the pathways that were most significantly represented among the differentially expressed genes. Ingenuity Pathway Analysis (IPA) revealed that integrin-linked kinase (ILK) signaling was the top deregulated pathway (Fig. 1e). Gene set enrichment analysis (GSEA) in chemotherapy-treated TNBC patients demonstrated that genes involved in focal adhesion signaling were significantly enriched in high DoxoR-GS scorers (Fig. 1f), underlining the importance of integrins and downstream focal adhesion in doxorubicin resistance.

Based on the RNA-seq analysis of our doxorubicin-resistant models, three integrin genes, *ITGA5*, *ITGA10* and *ITGB5* were significantly (FC cut-off = 1.5, $p$-value < 0.05) upregulated in resistant tumors (Fig. 1g and Supplementary Fig. 1c). We found that higher expression of only the integrin alpha 5 (ITGA5), but

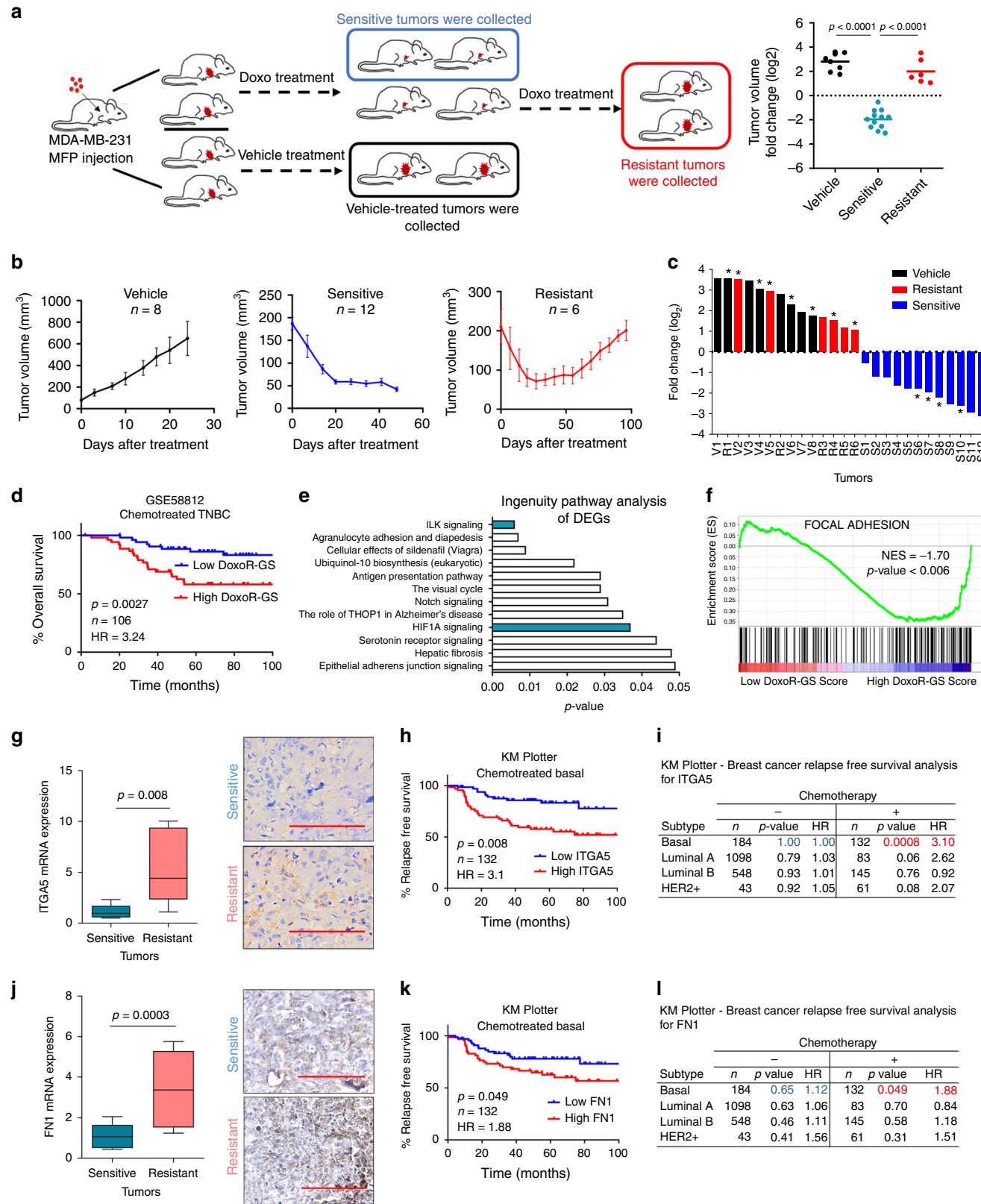

not that of ITGA10 or ITGB5, is associated with shorter relapse-free survival (RFS) in chemotherapy-treated basal patients (Fig. 1h, i and Supplementary Fig. 1d, e). No such relationship was observed in patients with basal subtype of breast cancer who were not treated with chemotherapy or in patients with other subtypes (Fig. 1i). As around 70% of TNBCs are basal subtype,

and 76% of basal subtype are TNBCs[38], we classified patients also as TNBC using ERα, PR, and HER2 expression and again observed a strong association between ITGA5 expression and RFS in chemotherapy-treated TNBC patients (Supplementary Fig. 2a). Importantly, we also detected a significant upregulation of fibronectin (FN1), the main ligand for ITGA5, in doxorubicin-

**Fig. 1 Integrin signaling is a key mediator of chemoresistance in TNBCs. a** Schematic representation of developing doxorubicin resistance in mice using the TNBC cell line, MDA-MB-231 (left panel). Tumor volume fold change (log2) of vehicle-treated, doxorubicin-sensitive and -resistant tumors (right panel). Clipart reprinted with permission from Springer Nature, *Nature Protocols*, FACS isolation of endothelial cells and pericytes from mouse brain microregions, Elizabeth E Crouch and Fiona Doetsch, Copyright 2018. **b** Tumor volumes of vehicle ($n = 8$), sensitive ($n = 12$) and -resistant ($n = 6$) tumors. **c** Waterfall plot of tumor volume fold change over time. Asterisk indicates tumors profiled by RNA-Seq. V vehicle; S sensitive; and R resistant. **d** Kaplan–Meier survival curve representing the percentage overall survival (OS) in chemotherapy-treated TNBC patients ($n = 106$) based on low vs. high (median) DoxoR-GS score. **e** Summary of IPA analysis showing top deregulated pathways in doxorubicin-resistant xenografts. **f** Genes associated with focal adhesion signaling are enriched in tumors of high DoxoR-GS scorers from GSE58812. **g** Expression of integrin alpha 5 (ITGA5) in doxorubicin-sensitive ($n = 8$) vs. -resistant ($n = 6$) xenografts at mRNA (left) and protein (right) levels, demonstrating membranous and cytoplasmic staining. **h** Kaplan–Meier survival curve representing the percentage relapse-free survival (RFS) in chemotherapy-treated basal patients ($n = 132$) based on low vs. high ITGA5 (median) expression. **i** Table summarizing association of ITGA5 with survival in patients of different subtypes that received or did not receive chemotherapy. **j** Expression of fibronectin (FN1) in doxorubicin-sensitive ($n = 12$) vs. resistant ($n = 12$) tumors at mRNA (left) and protein (right) levels, demonstrating mild to moderate cytoplasmic staining. **k** Kaplan–Meier survival curve representing the percentage RFS in chemotherapy-treated basal breast cancer patients ($n = 132$) based on low vs. high FN1 (median) expression. **l** Table summarizing association of FN1 with survival in patients of different subtypes that received or did not receive chemotherapy. Data on **b** represent mean ± SEM and all others represent mean ± SD. In Box plots, the box depicts median, 25th to 75th percentiles, and the whisker depicts min to max for this figure and all others. Two-sided Student's *t*-test was used to calculate statistical difference between two groups. Significance for survival analyses was calculated by log-rank (Mantel-Cox) test. NES Normalized enrichment score, HR hazard ratio. Scale bar = 100 µm for **g**, **j**. Source data are provided as a Source data file.

resistant tumors (Fig. 1j) and a significant association between high FN1 expression and poorer RFS in chemotherapy-treated basal patients (Fig. 1k, l). These results were further validated with another TNBC patient dataset (GSE58812) in which higher expression of either ITGA5 or FN1 is significantly associated with worse overall survival for the chemotherapy-treated TNBC patients (Supplementary Fig. 2b). Notably, when the patients are stratified based on ITGA5 and FN1 expressions in combination, a stronger separation of overall survival was achieved (Supplementary Fig. 2c, $p = 0.0008$; HR = 5.28). Altogether, these data suggest that deregulation of FN1-ITGA5 signaling could be a major driver of chemoresistance in TNBC.

**Hypoxia-induced LOX hyperactivates ITGA5/FN1/FAK/Src axis.** Hypoxia inducible factor-1 alpha (HIF-1α), encoded by the *HIF1A* gene, activates the transcription of several ECM-remodeling enzymes, including collagen prolyl and lysyl hydroxylases and lysyl oxidases thereby modulating ECM stiffness[39,40]. Importantly, our IPA analysis revealed a significant enrichment of HIF1A signaling in the doxorubicin-resistant tumors (Fig. 1e). Given the involvement of hypoxia in ECM remodeling[41,42], we hypothesized that HIF-1α could activate integrin and focal adhesion signaling. We first validated activation of the hypoxic response in chemoresistant tumors by demonstrating upregulation of the CA9 gene, which is a direct HIF-1α target gene and a well-established hypoxia marker[43]. CA9 mRNA and protein levels were significantly higher in resistant tumors (Fig. 2a). The induction of hypoxia signaling in the resistant tumors was not simply a result of an increase in tumor size, as there was no enrichment of hypoxia signaling in vehicle-treated tumors that are the largest in size vs. sensitive tumors (Supplementary Table 1). Furthermore, patients having high DoxoR-GS score also express high levels of hypoxia-related genes (Fig. 2b).

To experimentally demonstrate the acquisition of doxorubicin resistance under hypoxic conditions, we cultured MDA-MB-231 and MDA-MB-157 cells in normoxic vs. hypoxic conditions in the presence of increasing concentrations of doxorubicin. As shown in Fig. 2c, d, cells became less sensitive to growth inhibition induced by doxorubicin when grown under hypoxic conditions. Next, we performed Upstream Regulator analysis in IPA and found that 28 of 39 HIF-1α target genes had an expression direction consistent with the activation of HIF-1α (Supplementary Table 2). Among the actively transcribed genes upon HIF-1α activation during chemoresistance, we identified LOX, which is known to modulate ECM stiffness via collagen

cross-linking and promote metastasis[44]. Upregulation of LOX in doxorubicin-resistant xenografts was validated by qRT-PCR and immunohistochemistry (Fig. 2e). While LOX, ITGA5, and FN1 mRNAs were upregulated between sensitive and resistant tumors under doxorubicin treatment, the fast-growing vehicle-treated tumors showed no upregulation of these genes, suggesting that their upregulation is specific to chemotherapy resistance (Supplementary Fig. 3a). Importantly, we demonstrated increased fibrillar collagen in resistant tumors compared to sensitive counterparts by picrosirius red staining (Fig. 2f, g). We then examined gene expression from 9 different breast cancer datasets and found correlation between *HIF1A* and *LOX*, *ITGA5*, and *FN1* mRNAs (Fig. 2h), supporting the upstream regulatory role of HIF1A in their transcription. Strikingly, the correlation of *LOX* with *ITGA5* and *FN1* mRNAs was the strongest among all pairs, even stronger than the correlation of these three genes with *HIF1A*. This suggested that hypoxia-induced LOX might specifically be regulating *ITGA5* and *FN1* expression, and the subsequent activation of intracellular downstream signaling could be contributing to doxorubicin resistance. Consistent with this, we detected a significant enrichment of hypoxia and focal adhesion signaling gene sets in tumors with high LOX expression (GSE58812[36], Supplementary Fig. 4a, b).

To test whether hypoxia can induce both LOX expression and integrin signaling, we cultured MDA-MB-231 cells under hypoxia for different time points and observed a prominent increase in HIF-1α protein stability that was followed by a coordinated upregulation of LOX, ITGA5 and FN1 mRNAs and protein levels (Fig. 2i, j). Hypoxia also resulted in activation of integrin signaling as shown by incases in p-FAK (Y397) and p-Src (Y416) (Fig. 2j). Moreover, LOX enzymatic activity was higher under hypoxia as compared to normoxia, potentially due to increased LOX expression (Fig. 2k). Here, BAPN, a LOX family inhibitor, was used as a negative control. The induction of LOX/ITGA5/FN1 and downstream signaling under hypoxic conditions has also been validated in another TNBC cell line, MDA-MB-157 (Supplementary Fig. 4c). Silencing LOX expression using two different siRNA sequences caused a LOX expression-dependent decrease in both ITGA5 and FN1 mRNA (Fig. 2l) and protein levels (Fig. 2m) together with attenuated downstream signaling (Fig. 2m). Moreover, stable and inducible knockdown of LOX with shRNAs also validated the inhibition of downstream signaling (Fig. 2n). These data strongly support a model in which LOX-mediated upregulation of ITGA5 and its ligand, FN1 play a role in hypoxia-mediated hyperactivation of integrin signaling.

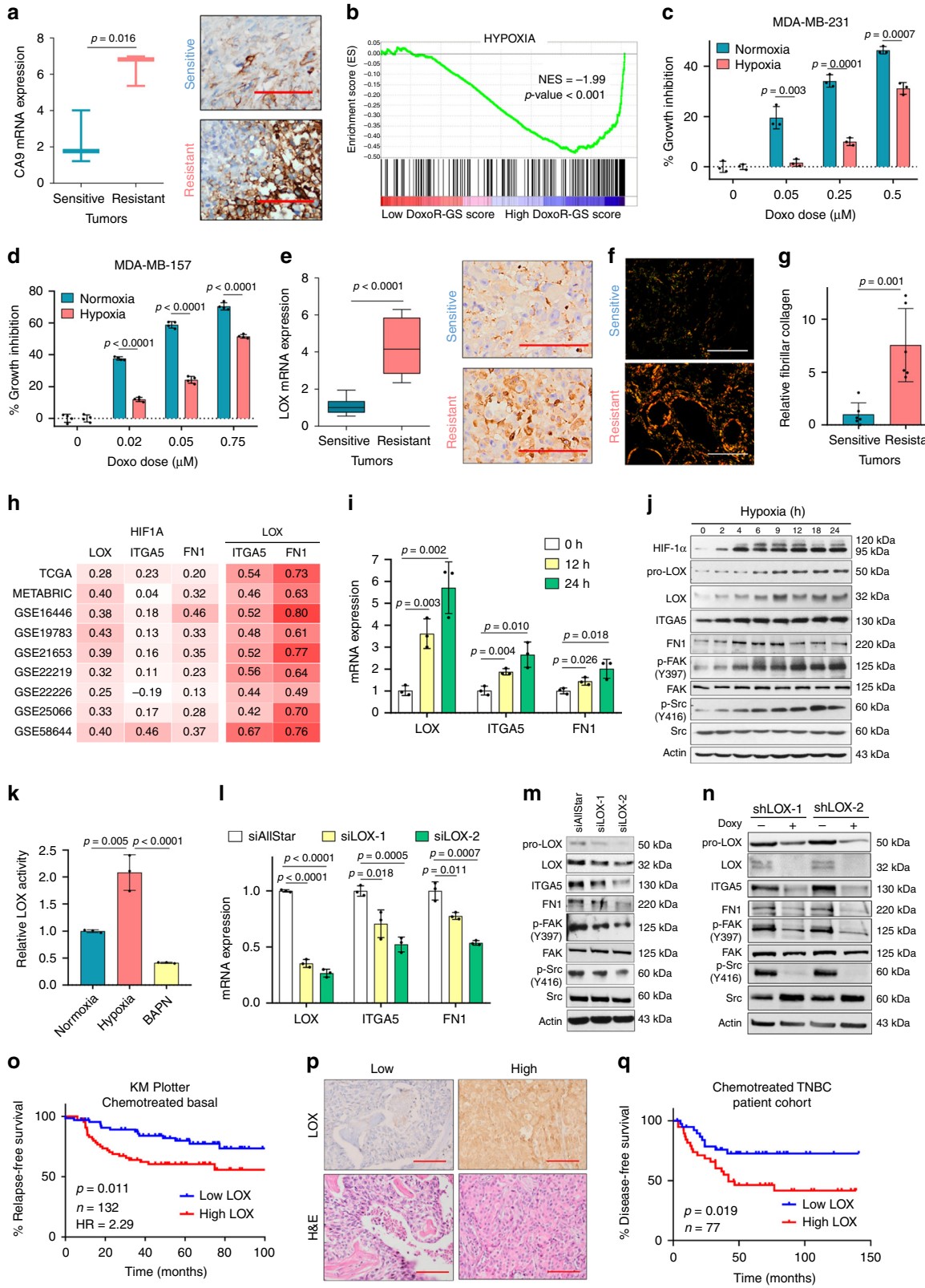

To determine the clinical relevance of LOX expression in chemotherapy-treated TNBCs, we performed survival analyses and observed that higher *LOX* mRNA expression predicts poor RFS only in chemotherapy-treated basal breast cancer patients, but not in other breast cancer subtypes or in untreated cases (Fig. 2o and Supplementary Fig. 2d). Furthermore, in the GSE31519 dataset we observed that more than half of the

patients that exhibit event, described as either a relapse or distant metastasis, express higher LOX levels ($p$-value = 0.012) (Supplementary Fig. 5a). Similarly, in another dataset (GSE25066), we observed that a significantly higher proportion of chemotherapy-treated TNBC patients with residual disease (RD) express higher levels of LOX ($p$-value = 0.011) (Supplementary Fig. 5b). Furthermore, TNBC patients with RD express significantly higher levels

**Fig. 2 Hypoxia-induced LOX hyperactivates ITGA5/FN1/FAK/Src axis in TNBCs. a** Expression of a HIF-1α direct target gene, carbonic anhydrase 9 (CA9) in sensitive ($n = 3$) vs. resistant ($n = 3$) tumors at mRNA (left) and protein (right) levels, demonstrating predominantly membranous and mild cytoplasmic staining. **b** Genes upregulated upon hypoxia are enriched in patients with high DoxoR-GS score from GSE58812 ($n = 106$). **c, d**. % growth inhibition upon doxorubicin treatment of MDA-MB-231 (**c**) ($n = 3$) and MDA-MB-157 (**d**) ($n = 4$) cells grown under normoxic vs. hypoxic conditions. **e** Expression of LOX in sensitive ($n = 12$) vs. resistant ($n = 9$) tumors at mRNA (left) and protein (right) levels, demonstrating strong cytoplasmic and weak nuclear staining. **f, g** Representative images of Picrosirius red staining (**f**) and its quantification (**g**) ($n = 6$). **h** Heatmap summarizing the Pearson's correlation coefficients between *HIF1A* and *LOX*, *ITGA5* or *FN1* and between *LOX* and *ITGA5* or *FN1* in breast cancer patients. An intense red color shows a stronger positive correlation. **i** qRT-PCR analysis of *LOX*, *ITGA5*, and *FN1* under hypoxia ($n = 3$). **j** Western blot analyses of HIF-1α, LOX and integrin signaling members under hypoxia. **k** Relative LOX activity in MDA-MB-231 cells under hypoxia. BAPN was used as a negative control ($n = 3$). **l** qRT-PCR analysis of *LOX*, *ITGA5*, and *FN1* after transfection with siAllStar or siLOX ($n = 3$). **m, n** Western blot analyses in MDA-MB-231 cells transfected with siAllStar or siLOX (**m**) and after shLOX induction with doxycycline (**n**). **o** Kaplan–Meier survival curve representing the percentage RFS in chemotherapy-treated basal breast cancer patients ($n = 132$) separated based on low vs. high (median) LOX mRNA. **p** IHC images of TNBC patient tissues with low and high LOX protein expression. **q** Kaplan–Meier survival curve representing DFS in chemotherapy-treated TNBC patients ($n = 77$) separated from median based on LOX protein expression. Data represents mean ± SD. Two-sided Student's *t*-test was used to calculate statistical difference between two groups. Significance for survival analyses was calculated by log-rank (Mantel-Cox) test. NES Normalized enrichment score, HR hazard ratio. Scale bar = 100 μm for **a**, **e**, and **p**, and 400 μm for **f**. Source data are provided as a Source data file.

of LOX, ITGA5 and FN1 (Supplementary Fig. 5c–e). Importantly, we stained for LOX protein expression in tissues (Fig. 2p, Supplementary Table 3). In 77 TNBC patients treated with chemotherapy, we showed that higher LOX protein levels were significantly associated with worse disease-free survival (DFS) (*p*-value = 0.019, Fig. 2q). Although other LOX family members (LOXL1-4) have also been reported to be regulated by HIF-1α[45], none of them were significantly altered in doxorubicin-resistant xenografts (Supplementary Fig. 3b). Altogether, these data are consistent with a model in which hypoxia-induced LOX activates integrin signaling by upregulating ITGA5 and FN1, leading to poor survival and chemoresistance in TNBC patients.

**LOX inhibition remodels ECM to confer chemosensitization.** To test the potential role of LOX in mediating chemoresistance in TNBCs and elucidating the underlying mechanisms, we first embedded TNBC cells into a matrix of type I collagen, the major substrate of LOX, and demonstrated acquisition of resistance to doxorubicin in two different cell lines (Fig. 3a and Supplementary Fig. 4d) which was accompanied by protection against apoptosis (Fig. 3b and Supplementary Fig. 6a–c) and sustained levels of p-FAK and p-Src (Fig. 3c). Western blot analysis also demonstrated a decrease in HIF-1α expression and subsequently LOX levels upon doxorubicin treatment but only in cells cultured in the absence of collagen (Fig. 3c). Overexpressing LOX in MDA-MB-231 cells that were embedded in type I collagen conferred resistance to doxorubicin (Fig. 3d). Overexpression of LOX also caused a modest increase in ITGA5 protein levels (Fig. 3e). Analysis of LOX expression in a panel of breast cancer cell lines demonstrated that TNBC cells express relatively higher levels of LOX (Supplementary Fig. 7a). Importantly, inhibiting LOX using a LOX family inhibitor, BAPN in combination with doxorubicin in three different collagen I-embedded TNBC cell lines re-sensitized them to doxorubicin (Fig. 3f, Supplementary Fig. 7b). As BAPN is known to inhibit other LOX family members as well, we further confirmed the doxorubicin sensitization with LOX inhibition using two different siRNA sequences against LOX (Fig. 3g). Importantly, doxorubicin sensitization by LOX inhibition was specific to TNBC models. ERα+ cells express relatively low levels of LOX, and LOX inhibition did not lead to chemosensitization (Supplementary Fig. 7a, c). Finally, LOX inhibition sensitizes cells to two other DNA-damaging chemotherapy agents, epirubicin and paclitaxel (Supplementary Fig. 7d), which are common therapeutic options for TNBC patients[46].

To elucidate the mechanistic details of how LOX inhibition sensitizes TNBC cells to doxorubicin, we first examined the changes in ECM remodeling upon LOX inhibition. It has been reported that there is a molecular co-dependence between collagen cross-linking and fibronectin fibril assembly that causes reciprocal activation of both processes to ultimately increase ECM stiffness[47,48]. In addition, cell-mediated processes, such as binding of cell surface integrins to fibronectin dimers and subsequent recruitment of FAK to focal adhesion sites can trigger fibronectin fibrillogenesis[49–51]. We observed that both ITGA5 and FN1 expression, and subsequent FAK phosphorylation were dependent on LOX expression (Fig. 2). We therefore hypothesized that both collagen cross-linking (by LOX) and fibronectin deposition and assembly would be hindered by LOX inhibition. To test this hypothesis, we performed immunofluorescence staining of collagen and fibronectin in type I collagen-embedded MDA-MB-231 cells with and without LOX inhibition, and observed that TNBC cells are able to deposit large amounts of fibronectin into an ECM that is rich in collagen type I (Fig. 3h), and that LOX inhibition with BAPN decreased the extracellular deposition and assembly of fibronectin fibrils (Fig. 3h) and decreased overall extracellular fibrillar collagen content (Fig. 3i). This was validated using a cell-derived ECM system produced by human foreskin fibroblast (HFF) cells, better recapitulating the dynamics of ECM formation and stabilization. Growing cancer cells on fibroblast-derived ECM induced the formation of thick, cross-linked collagen fibers, as opposed to non-fibrillar, immature collagen in the ECM deposited in the absence of cancer cells (Fig. 3j). When LOX was inhibited with BAPN, we detected a significant decrease in the immunofluorescence staining intensities of both collagen and fibronectin, suggesting that cancer cell-derived LOX is required for the cross-linking of extracellular collagen into thick fibers and the subsequent fibronectin assembly during ECM maturation (Fig. 3k). Co-localization analysis demonstrated that almost half of the collagen fibers were in contact with fibronectin (similar to previous reports[47,48]), but co-localization was decreased by LOX inhibition (Supplementary Fig. 8a). Analysis of 3D images of vehicle and BAPN-treated cells cultured on top of HFF-derived ECM further demonstrated that cells are still in contact with the ECM even though LOX inhibition decreased the extracellular collagen and fibronectin content (Supplementary Fig. 8b, c). The decrease in collagen cross-linking and fibronectin assembly upon LOX inhibition was also validated by biochemical measurement of extracellular hydroxyproline content (Fig. 3l) and deoxycholate (DOC) lysis (Fig. 3m), respectively, which are standard methods used to assess the amount of collagen cross-linking and fibronectin deposition and assembly[52].

It has been proposed that LOX may reduce drug penetration under hypoxic conditions and lessen the cytotoxicity of chemo-agents in 3D collagen cultured cells and tumor models[53].

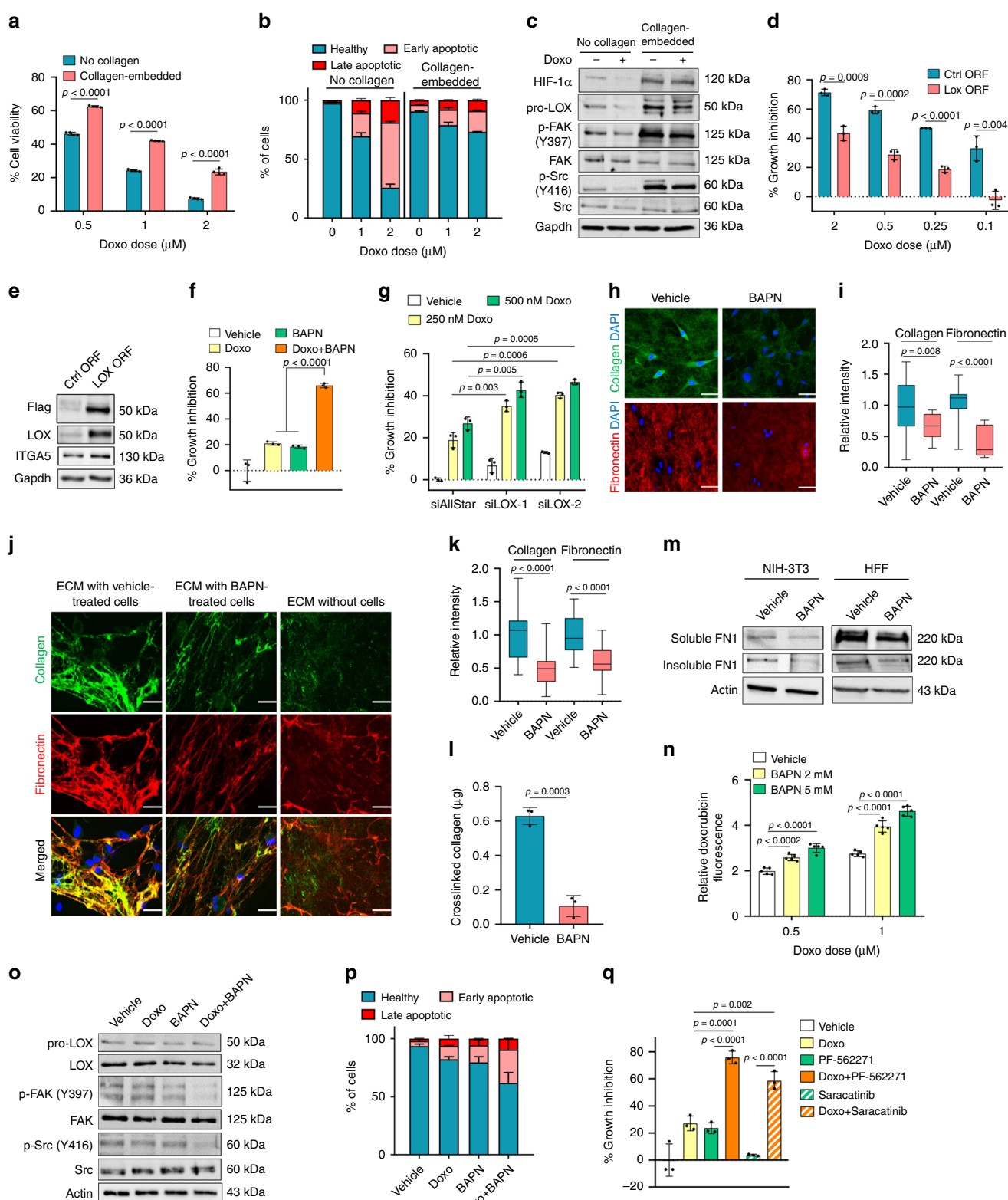

We showed that doxorubicin decreased the activation of FAK/Src signaling in 2D-cultured cells, but not in 3D collagen culture (Fig. 3c). Therefore, we tested if targeting LOX could increase doxorubicin penetration and decrease FAK/Src signaling, leading to apoptosis. To assess drug penetration, we quantified doxorubicin autofluorescence (ex: 495 nm, em: 595 nm[54]) and observed a significant increase in intracellular doxorubicin in combination-treated cells (Fig. 3n). This was accompanied by a decrease in FAK/Src phosphorylation (Fig. 3o) and induction of apoptosis (Fig. 3p and Supplementary Fig. 6d–g). Finally, we treated collagen-embedded cells with doxorubicin with and without inhibitors specific for FAK (PF-562271) or Src (Saracatinib) and observed a synergistic increase in growth inhibition (Fig. 3q), decreased phosphorylation of FAK and Src (Supplementary Fig. 9a), and induced apoptotic cell death (Supplementary Fig. 9b, c).

**Fig. 3 LOX inhibition remodels ECM to confer chemosensitization in TNBCs. a** Doxorubicin response of MDA-MB-231 cells cultured with or without type I collagen for 72 h ($n = 4$). **b** Apoptosis assay by Annexin V/DAPI staining from a ($n = 2$). **c** Western blot analysis in doxorubicin-treated cells grown with or without type I collagen. **d** Percentage growth inhibition in LOX-overexpressing cells embedded in collagen upon doxorubicin treatment ($n = 3$). **e** Western blot analyses upon LOX overexpression in MDA-MB-231 cells. **f, g** Percentage growth inhibition of collagen-embedded MDA-MB-231 cells treated with BAPN (**f**) or transfected with siLOX (**g**) in combination with doxorubicin ($n = 3$). **h, i** Immunofluorescence staining (**h**) and quantifications of the intensities (**i**) of extracellular type I collagen and fibronectin upon treatment of collagen-embedded cells with BAPN ($n = 18$ (vehicle), $n = 15$ (BAPN) for collagen, and $n = 18$ (vehicle), $n = 12$ (BAPN) for fibronectin). **j, k** Immunofluorescence staining (**j**) and quantifications of the intensities (**k**) of HFF-derived type I collagen and fibronectin incubated with vehicle or BAPN-treated MDA-MB-231 cells. ECM without cells represents the staining in the absence of MDA-MB-231 cells ($n = 29$ (vehicle), $n = 25$ (BAPN)). **l** Amount of cross-linked collagen in HFF-derived ECM incubated with vehicle- vs. BAPN-treated MDA-MB-231 cells ($n = 3$ different wells). **m** Western blot of soluble and insoluble FN1 obtained by deoxycholate lysis of NIH3T3- and HFF-derived ECM in contact with vehicle- vs. BAPN-treated MDA-MB-231 cells. **n** Changes in relative doxorubicin fluorescence upon BAPN-treatment in MDA-MB-231 cells embedded in type I collagen ($n = 5$). **o** Western blot analysis of LOX and FAK/Src signaling in collagen type I-embedded MDA-MB-231 cells upon doxorubicin and BAPN treatment for 24 h. **p** Annexin V/DAPI staining upon combination treatment for 72 h ($n = 2$). **q** Percentage growth inhibition induced by the combination of doxorubicin with FAK (PF-562271) or Src (Saracatinib) inhibitors in MDA-MB-231 cells embedded in type I collagen ($n = 3$). Data represents mean ± SD. Two-sided Student's *t*-test was used to calculate statistical difference between two groups. One-way ANOVA with Dunnett's test was performed to compare mean of combination-treated group with single agent treatments in **f, q**. Scale bar = 50 μm for **h, j**. Source data are provided as a Source data file.

**Targeting LOX or FAK/Src overcomes chemoresistance in vivo.** To test if inhibiting LOX could overcome doxorubicin resistance and enhance drug response in advanced, chemotherapy-refractory TNBCs, we re-derived doxorubicin resistance in vivo, using a protocol similar to our initial acquired resistance model (though with lower doxorubicin dose), by treating MDA-MB-231 xenografts with doxorubicin until an accelerated tumor growth was achieved despite the given therapy. At this point, some of the doxorubicin-resistant tumors were treated with the combination of doxorubicin and BAPN while the rest continued to receive doxorubicin alone. Strikingly, addition of LOX inhibitor to doxorubicin in these aggressive, fast-growing doxorubicin-resistant tumors led to a significant decrease in tumor growth rate (Fig. 4a, b) and improved survival (Fig. 4c) until experiment termination when the tumor volume cut-off was reached or due to almost a 20% decrease in body weight in BAPN-added treatment arm (Fig. 4d). We confirmed the reduction of LOX activity (Fig. 4e) and fibrillar collagen (Fig. 4f, g) upon addition of BAPN to doxorubicin. These results were further validated by performing biochemical collagen assay (Fig. 4h), suggesting that chemosensitization with LOX inhibition involves reduction in collagen cross-linking.

We also evaluated the effects of LOX inhibition on doxorubicin response in vivo in an immunocompetent setting using a syngeneic TNBC model. The 4T1 mouse mammary tumor cell line is a highly aggressive model that represents the TNBC subtype[55] and exhibits relatively low response to doxorubicin. We examined the expression of LOX in 4T1 cells grown in vitro as well as in tumors and found that they express LOX at a level similar to the high LOX-expressing MDA-MB-231 model (Supplementary Fig. 10a, b). We observed a strong inhibitory effect of the combination therapy on tumor growth in this immunocompetent tumor model (Fig. 4i and Supplementary Fig. 10c, d). The LOX activity assay and picrosirius red staining confirmed the suppression of LOX activity and collagen cross-linking, respectively in all BAPN-treated tumors (Fig. 4j–l). This was accompanied by increased penetration of doxorubicin (Fig. 4m, n) and inhibition of FAK/Src signaling in the combination-treated tumors (Fig. 4o). To test the effect of FAK or Src inhibition on doxorubicin response in vivo, we combined specific inhibitors of FAK (PF-562271) or Src (Saracatinib) with doxorubicin and observed a significant decrease in tumor growth (Fig. 4p–s) and tumor weight (Supplementary Fig. 10e, f) in combination-treated tumors, validating the key roles of FAK and Src in doxorubicin resistance.

Next, we tested combination of LOX inhibition with doxorubicin in a highly aggressive TNBC PDX model. We first analyzed the gene expression profiling and drug response data of

15 well-characterized TNBC PDX models (Jackson Lab) and observed a significant positive correlation between LOX expression and hypoxia and focal adhesion scores (Fig. 5a). Based on these analyses, we selected a TNBC PDX model (TM01278) that is resistant to doxorubicin (and several other chemotherapy agents) and expresses high levels of LOX, hypoxia and focal adhesion scores (Fig. 5a). Then, we generated organoid cultures of this doxorubicin-resistant PDX tumor and demonstrated that combination of LOX inhibitor with doxorubicin significantly decreased organoid size compared to single agent treatments after 9 days of treatment (Fig. 5b, c). In vivo testing also showed a significant decrease in tumor growth upon combination therapy in this chemoresistant PDX model expressing LOX (Fig. 5d–f). These results complement our in vitro and in vivo findings with established cell lines and increase the clinical relevance of LOX inhibition to overcome chemotherapy resistance in TNBC. Importantly, we demonstrated reduced LOX activity (Fig. 5g) and fibrillar collagen content upon LOX inhibition with BAPN (Fig. 5h, i), which was accompanied by enhanced drug penetration into tumors (Fig. 5j, k) and more effective downregulation of downstream FAK/Src signaling in combination-treated tumors (Fig. 5l). Overall, we demonstrated that targeting LOX or its downstream FAK/Src overcomes chemotherapy resistance in highly aggressive, chemoresistant TNBC models.

**Targeting LOX at the first-line potentiates chemoresponse.** Having identified LOX as a determinant of chemoresistance in TNBC, we tested whether LOX is also a target in potentiating chemotherapy response in first-line settings in treatment-naïve models using our inducible shLOX expressing MDA-MB-231 derivatives (referred to 231.shLOX) (Supplementary Fig. 11a, b). Inhibiting LOX in combination with doxorubicin as a first-line therapy led to a stronger delay in tumor growth as compared to individual treatments (Fig. 6a–e). We confirmed LOX knockdown at mRNA (Fig. 6f) and protein levels (Fig. 6g, h) and the decrease in LOX activity (Supplementary Fig. 11c) together with downregulation of its downstream ITGA5 (Supplementary Fig. 11d) after tumors were collected. We also demonstrated that other LOX family members did not decrease upon LOX knockdown (Supplementary Fig. 11e, f), further supporting our hypothesis that LOX inhibition is specifically involved in mediating chemoresponse in TNBCs. In consistent with the reduction in tumor growth, there was also a significant reduction in the proliferation marker, Ki-67 (Fig. 6i, j) and an increase in the expression of an apoptosis marker, Cleaved Caspase-3 (Fig. 6k, l) in combination-

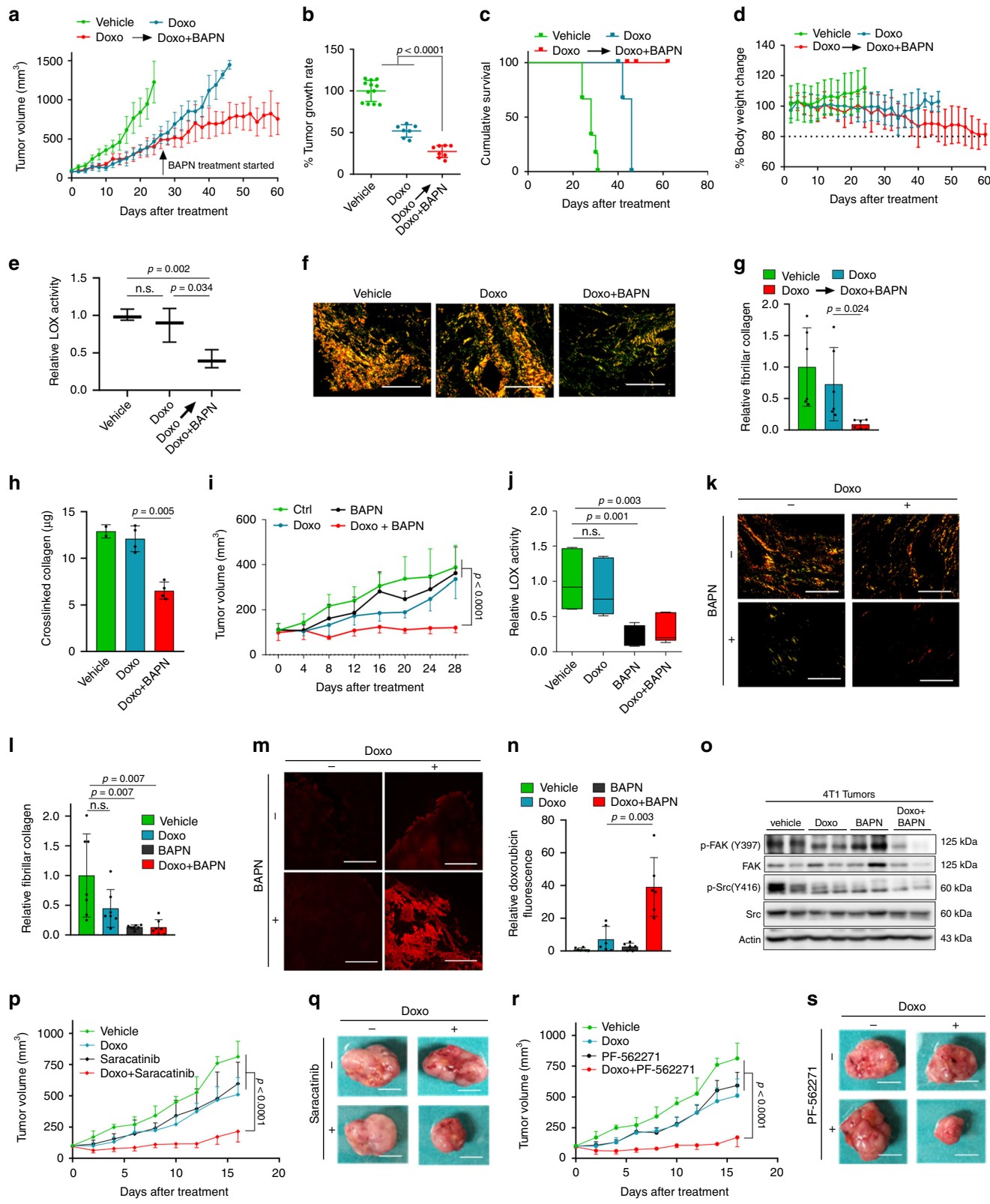

treated group as compared to LOX inhibition or doxorubicin treatment alone. Furthermore, ITGA5 levels, phosphorylation of Src (p-Y416) and FAK (p-Y397) were reduced upon LOX knockdown in combination with doxorubicin (Fig. 6m–q).

Next, we examined the effect of combination therapy on the growth of a primary organoid model developed from a treatment-naïve TNBC patient. Combination treatment with doxorubicin and BAPN led to a significant reduction in organoid size as compared to individual treatment groups after 9 days of treatment (Fig. 6r, s). Altogether, these data demonstrate that inhibition of LOX could be a potential therapeutic strategy in combination with first-line chemotherapy in TNBCs.

**Fig. 4 Targeting LOX or downstream FAK/Src overcomes TNBC chemoresistance in vivo. a** Tumor growth in MDA-MB-231 xenografts treated with doxorubicin until resistance develops followed by treatment with the combination of doxorubicin (2.5 mg/kg) and the LOX inhibitor, BAPN (100 mg/kg) ($n = 12$, 7, and 8 tumors for vehicle, Doxo and Doxo+BAPN, respectively). **b** Tumor growth rates relative to vehicle from **a**. **c** Cumulative survival of mice from **a**. Mice were sacrificed when the tumor size cut-off was reached or when the body weight dropped to 80% of the initial body weight ($n = 6$, 4, and 4 mice for vehicle, Doxo and Doxo + BAPN, respectively). **d** Percentage change in the body weight of the mice from **c**. **e** Relative LOX activity in tumors from **a** ($n = 3$). **f** Representative images of Picrosirius red staining (f) and its quantification (**g**), in tumors from **a** ($n = 6$). **h** In vivo collagen assay in tumors from **a** ($n = 2$ (vehicle), $n = 4$ (Doxo, Doxo+BAPN)). **i** Tumor growth in 4T1 syngeneic model upon treatment with doxorubicin and BAPN, alone or in combination ($n = 4$ mice). **j** LOX activity in tumors from **i** ($n = 6$). **k**, **l** Representative images of Picrosirius red staining (k) and its quantification (l) in mice from i ($n = 7$). **m**, **n** Intratumoral doxorubicin levels upon treatment with the combination of doxorubicin and BAPN ($n = 6$). **o** Western blot analysis of FAK/Src signaling in combination-treated tumors from **i**. **p**, **q** Change in tumor growth (**p**) and representative images of tumors (**q**) in 4T1 model upon treatment with doxorubicin and Saracatinib, alone or in combination ($n = 4$). **r**, **s** Change in tumor growth (**r**) and representative images of tumors (**s**) in 4T1 model upon treatment with doxorubicin and PF-562271, alone or in combination ($n = 4$). Data represents mean ± SD. Two-sided Student's $t$-test was used to calculate statistical difference between two groups. Two-way ANOVA test was performed for comparing tumor growth over time among different treatment groups in **i**, **p**, and **r**. n.s. not significant. Scale bar = 400 μm for **f**, **k**; 200 μm for **m**, and 1 cm for **q**, **s**. Source data are provided as a Source data file.

**miR-142-3p is a chemosensitizer regulating HIF1A/LOX/ITGA5**. To elucidate the underlying molecular mechanisms of the coordinated activation of the HIF1A/LOX/ITGA5 axis under hypoxia, we sought for a common modulator. As miRNAs tend to function in a pathway-centric manner by targeting multiple genes in the same cascade[56], we searched for a potential miRNA modulator of HIF1A/LOX/ITGA5-mediated chemotherapy resistance. We examined all conserved miRNAs targeting *HIF1A*, *LOX*, and *ITGA5* and found eight common miRNAs having binding sites in the 3′-UTRs of these three genes (Fig. 7a). Out of these eight miRNAs, only two miRNAs, miR-142-3p, and miR-128-3p, showed an inverse correlation with HIF1A signature, and LOX and ITGA5 expression in basal subtype (Fig. 7b and Supplementary Fig. 12a–c). One of these two miRNAs, miR-142-3p, showed a significant association with OS specifically in chemotherapy-treated TNBC patients (Fig. 7c, d and Supplementary Fig. 12d). Furthermore, miR-142-3p was significantly downregulated in doxorubicin-resistant xenografts that we developed in vivo as compared to sensitive ones (Fig. 7e). Importantly, induction of hypoxia inhibited the expression of miR-142-3p (Fig. 7f), while silencing HIF1A upregulated miR-142-3p expression (Fig. 7g), suggesting that hypoxia-mediated downregulation of miR-142-3p potentially involves HIF1A.

We next tested to see if miR-142-3p could inhibit the expression of *HIF1A*, *LOX* or *ITGA5*. We overexpressed miR-142-3p in MDA-MB-231 cells and observed a significant downregulation of all three genes both at mRNA and protein levels, and a reduction in phosphorylation of downstream FAK/Src kinases (Fig. 7h, i). Importantly, downregulation of HIF1A, LOX and ITGA5 protein levels as well as decreased phosphorylation of FAK and Src was also seen in MDA-MB-231 xenografts stably expressing miR-142-3p (Fig. 7j and Supplementary Fig. 12e). The direct binding of miR-142-3p to the predicted binding sites in the 3′-UTRs of *HIF1A*, *LOX*, and *ITGA5* (Fig. 7k) was validated by a dual luciferase assay (Fig. 7l). Finally, ectopic expression of miR-142-3p in TNBC cells grown in type I collagen sensitized to doxorubicin-induced growth inhibition and apoptotic cell death (Fig. 7m–p). We interpret these results to indicate that HIF1A and miR-142-3p are involved in a double-negative feedback loop that further increases HIF1A, LOX and ITGA5 levels under hypoxic conditions, leading to chemoresistant tumors, and that overexpression of miR-142-3p overcomes chemoresistance by inhibiting HIF1A/LOX/ITGA5 axis.

## Discussion

TNBC is the most aggressive subtype of breast cancer and is responsible for 30% of all breast cancer deaths. Chemotherapy is the mainstay treatment for TNBCs; however, resistance is

common, significantly decreasing long-term survival[6]. Therefore, novel strategies are urgently needed to enhance the clinical benefit from chemotherapy. Here, we identified a mechanism of chemotherapy resistance that involves activation of the HIF1A/miR-142-3p/LOX/ITGA5/FN1 axis in TNBCs. We showed that inhibiting LOX reduces the expression of ITGA5 and FN1, decreases extracellular collagen cross-linking, fibronectin deposition/assembly and enhances drug penetration that ultimately result in inhibition of FAK/Src signaling, resulting in apoptosis and reversal of chemotherapy resistance in TNBC cells cultured in contact with ECM (Fig. 3). These results were validated in acquired resistant TNBC xenografts and de novo resistant 4T1 syngeneic TNBC model (Fig. 4) as well as in well-characterized chemoresistant TNBC PDXs (Fig. 5). We also showed that targeting LOX potentiates chemotherapy in first-line settings (Fig. 6). Importantly, targeting the signaling molecules downstream of LOX, FAK or Src, strongly sensitized cancer cells to chemotherapy when grown in collagen-embedded cultures or in vivo, showing the key role of these pro-survival signals in driving resistance (Fig. 4). Finally, we showed a double-negative feedback loop between HIF1A and miR-142-3p regulating HIF1A/LOX/ITGA5/FAK/Src axis and chemoresistance in TNBC (Fig. 7).

The ECM re-modeler, LOX is a well-known modulator of cancer metastasis[29,44]. LOX secretion from hypoxic tumors leads to collagen cross-linking at the pre-metastatic site and increases lung metastasis in TNBC models[28]. Moreover, higher LOX expression was observed in metastatic brain tumors of breast cancer patients[57]. Furthermore, Baker et al. demonstrated that LOX increases cell proliferation and invasion in colorectal cancer via Src activation[58] and enhances matrix stiffness, activates FAK, leading to invasion in vitro and metastasis in vivo in colorectal cancer[59]. In terms of the potential effects of LOX on drug response, a few studies have reported an involvement of LOX in drug resistance that primarily focused on altered distribution/penetration of chemotherapeutics, including paclitaxel and gemcitabine in tumors as a consequence of LOX-mediated ECM stiffness under hypoxic conditions[30,53]. In addition to forming a physical barrier, LOX-mediated ECM stiffness can also confer resistance via activating integrin receptors and the downstream effectors, FAK and Src kinases, leading to increased cellular survival[60]. For instance, Miller et al. demonstrated that combination of a LOX antibody with gemcitabine improved survival of a pancreatic ductal adenocarcinoma (PDAC) model via inhibiting key microenvironment-mediated pro-survival signals without any evidence of increased penetrance of gemcitabine into tumors upon LOX inhibition[31] in contrast to the studies described above[30,53]. Furthermore, it was recently shown that LOX might also be involved in transcriptional regulation of certain genes[61]. In our study, we showed that LOX

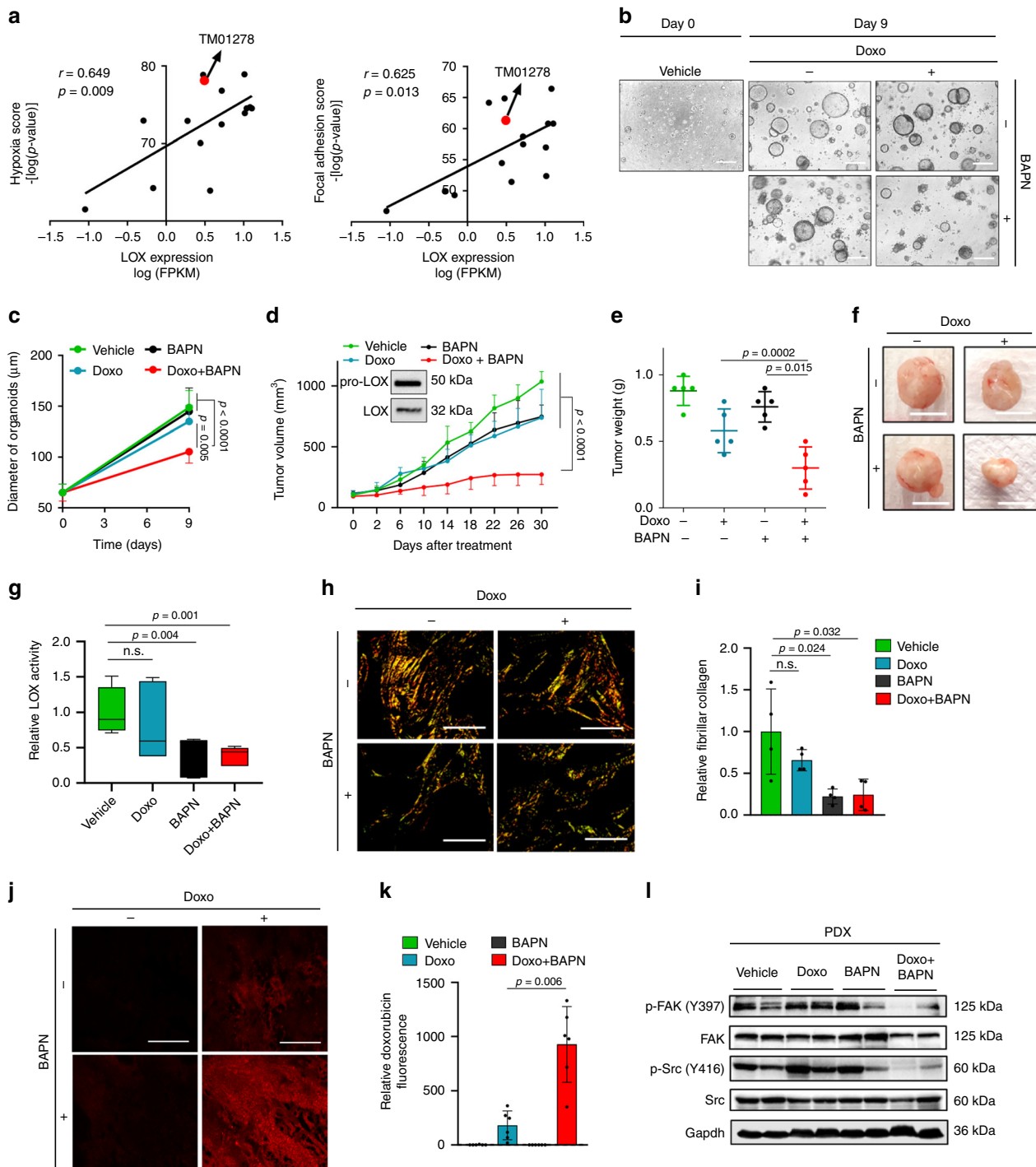

**Fig. 5 Targeting LOX overcomes chemoresistance in highly aggressive TNBC PDXs. a** Correlation analysis of *LOX* mRNA expression with hypoxia and focal adhesion scores in 15 different TNBC PDX models. Red dot shows the position of TM01278 PDX model selected. **b** Representative images of TM01278 PDX organoids at day 0 and day 9 after treatment with doxorubicin and BAPN treatment, alone or in combination. **c** Quantification of organoid diameter upon combination therapy for 9 days (*n* = 12 (vehicle, Doxo, BAPN), *n* = 11 (Doxo+BAPN)). **d, e** Tumor growth (**d**) and tumor weight (**e**) in TM01278 PDX upon treatment with doxorubicin and BAPN, alone or in combination (*n* = 5). Inset shows LOX expression in PDX tumors. **f** Representative images of tumors from **d**. **g** Relative LOX activity in tumors from **d** (*n* = 8 (vehicle), *n* = 6 (Doxo, BAPN, Doxo+BAPN)). **h, i** Representative images of Picrosirius red staining (**h**) and its quantification (**i**) in combination- and single agent-treated PDX tumors from d (*n* = 4). **j, k** Representative images of doxorubicin fluorescence in tumors from **d** (**j**), and its quantification (**k**) (*n* = 6). **l** Western blot analysis of FAK/Src signaling in PDXs treated with doxorubicin and BAPN, alone or in combination. Data represents mean ± SD. Two-sided Student's *t*-test was used to calculate statistical difference between two groups. One-way ANOVA with Dunnett's test was performed to compare mean of combination-treated group with single agent treatments in **e**. Two-way ANOVA test was performed for comparing tumor growth over time among different treatment groups in **d**. Scale bar = 100 μm for **b**, 1 cm for **f**, and 400 μm for **h** and 200 μm for **j**. n.s. not significant. Source data are provided as a Source data file.

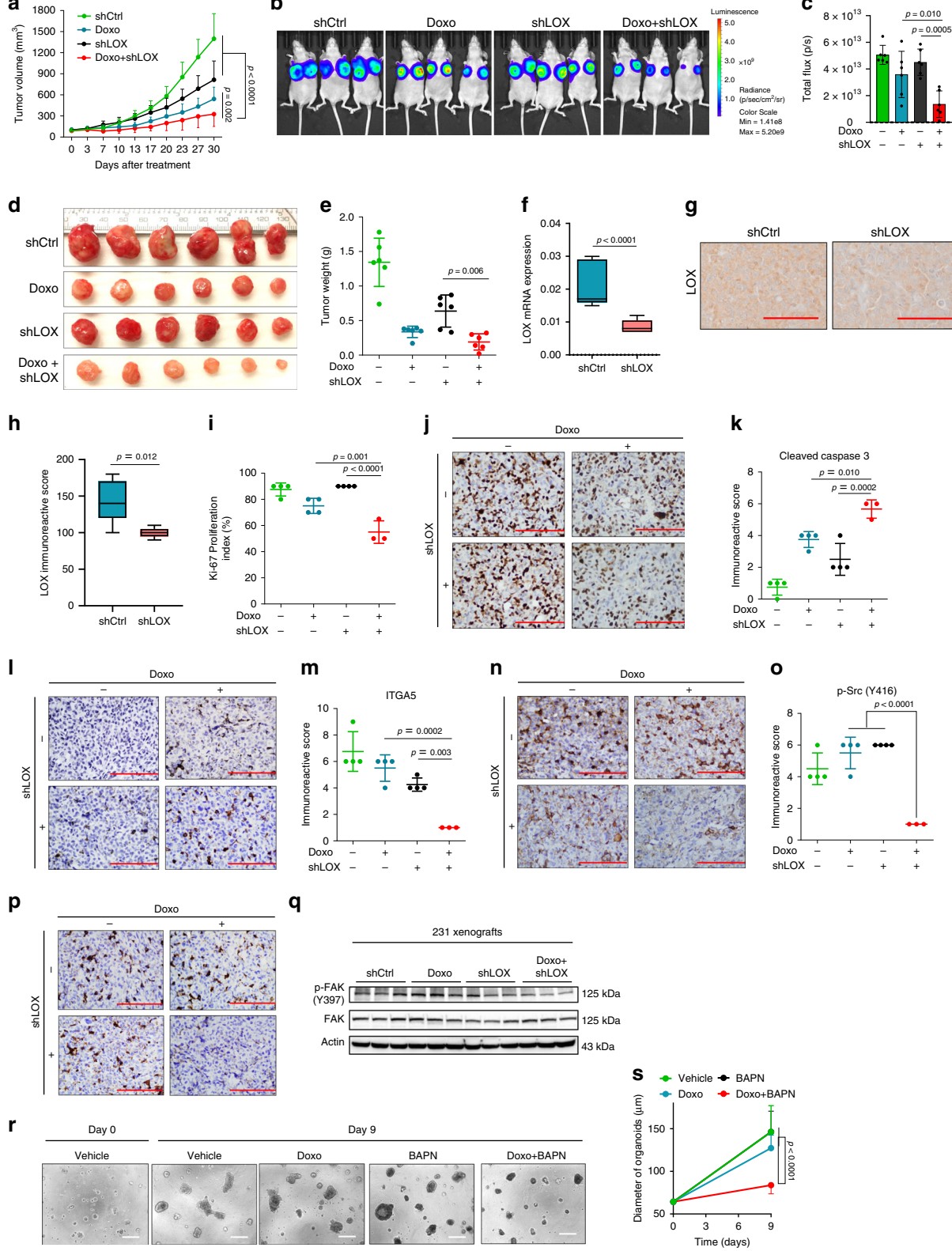

overexpression upon acquisition of doxorubicin resistance, on one hand, leads to protection against doxorubicin-induced FAK/Src inhibition by increasing collagen cross-linking, fibronectin assembly and decreasing drug penetration, and on the other hand, increases ITGA5 and FN1 expression, ultimately leading to increased FAK/Src signaling and chemoresistance. Considering the known potential of inhibiting FAK/Src kinases

to enhance response to different chemotherapy agents[62] and our data on the effect of LOX inhibition on FAK/Src signaling in chemosensitization, we propose that LOX inhibition reduces both the ECM stiffness to enhance drug penetration and the ITGA5/FN1 expression, thus culminating in inhibition of FAK/Src signaling, induction of apoptosis and chemosensitization. Overall, these findings suggest that LOX,

**Fig. 6 Targeting LOX at the first-line settings potentiates chemoresponse in TNBCs. a** Tumor volume in MDA-MB-231 xenografts upon shLOX induction in the presence or absence of doxorubicin treatment ($n = 6$). **b** IVIS images of mice from **a**. **c** Quantifications of luciferase intensity in tumors from **b** ($n = 6$). **d** Images showing isolated tumors from **a**. **e** Tumor weights in mice from **a** ($n = 6$). **f** qRT-PCR analysis of LOX in Ctrl ($n = 9$) vs. shLOX ($n = 9$) xenografts. **g, h** IHC staining of LOX and its quantification in Ctrl ($n = 5$) vs. shLOX ($n = 5$) tumors. **i, j** Ki-67 proliferation index of tumors from a ($n = 4$ (vehicle, Doxo, BAPN), $n = 3$ (Doxo+BAPN)). **k–p** Immunoreactive scores of Cleaved Caspase-3 (**k, l**), ITGA5 (**m, n**) and p-Src (Y416) (**o, p**) in tumors from a ($n = 4$ (vehicle, Doxo, BAPN), $n = 3$ (Doxo + BAPN)). **q** Western blot analysis of p-FAK and FAK in shLOX tumors in combination with doxorubicin. **r** Representative images of TNBC patient organoid, F149T at day 0 and day 9 after treatment with doxorubicin and BAPN, alone or in combination. **s** Quantification of organoid diameter upon combination therapy for 9 days ($n = 12$ (Day 0), $n = 9$ (vehicle, Day 9), $n = 11$ (Doxo, BAPN, Doxo+BAPN, Day 9)). Data represents mean ± SD. Two-sided Student's $t$-test was used to calculate statistical difference between two groups. One-way ANOVA with Dunnett's test was performed to compare mean of combination-treated group with single agent treatments in **c, e, i, k, m, o**, and **s**. Two-way ANOVA test was performed for comparing tumor growth over time among different treatment groups in **a**. Scale bar = 100 μm for **g, j, l, n, p, r**. Source data are provided as a Source data file.

which has primarily been studied in the context of metastasis, may also confer resistance to chemotherapy, making it a highly attractive therapeutic target.

Given the functional contribution of ECM modulation on different aspects of tumorigenesis, several inhibitors against multiple ECM modulators, such as matrix metalloproteinases, or the ECM-sensing integrin receptors have so far been developed. However, none of them achieved clinical success due to context-dependent efficacy, low specificity or severe toxicity[63]. Furthermore, inhibitors targeting the HIF pathway have so far been unable to enter clinics due to the complexity of the HIF pathway and the difficulty in targeting protein-protein interactions[64]. Therefore, identification of the well-defined modulators of ECM that are specifically over-expressed in aggressive, chemoresistant tumors, such as LOX that stands out as an attractive target with strong translational profile will be beneficial towards achieving a superior clinical outcome[44,65]. In this line, we demonstrated that more than half of the chemoresistant TNBC patients express high levels of LOX mRNA (Supplementary Fig. 5a, b) and are therefore expected to respond to LOX targeting therapies in combination with chemotherapy. In addition, we showed that high expression of LOX protein is significantly associated with survival in a cohort of chemotherapy-treated TNBC patients. Moreover, we also demonstrated a superior anti-tumorigenic effect of doxorubicin in vivo and in a TNBC organoid when combined with BAPN in the first-line settings (Fig. 6) that altogether supports the translational potential of targeting LOX with a potentially large target population in the clinic. Although BAPN is the most commonly used LOX inhibitor, it inhibits all LOX family members. In this line, there have been several recent efforts to identify novel and selective small molecule inhibitors against different family members, including LOX[66] and LOXL2[67] that can hopefully be tested in clinics to improve patient outcome in aggressive cancers, including the chemoresistant TNBCs. Moreover, our in vitro and in vivo data showing strong chemosensitization upon inhibiting FAK or Src kinases suggest that FAK/Src axis is critical for tumor cell survival in the presence of chemotherapy and targeting these proteins could also be an effective strategy to overcome chemoresistance in TNBCs in the future. Indeed, inhibitors of FAK and Src kinases are currently being tested in clinical trials and demonstrate promising results when combined with targeted therapies or chemotherapy (e.g. NCT03875820 and NCT02389309, respectively).

Signaling pathways need to be tightly regulated and are repressed in the absence of a stimulus. This ensures that target genes are activated only in the presence of a signal. miRNAs have been shown to be key regulators of this repression by inhibiting the expression of several transcripts[68]. Furthermore, others and us have previously shown that expression of genes functioning in the same cascade can be co-regulated by the same miRNA to ensure its robustness[56,68]. Therefore, we asked if miRNAs can inhibit the HIF1A/LOX/ITGA5 axis and confer chemosensitivity. We identified miR-142-3p to be significantly downregulated in chemoresistant tumors by HIF1A, and its overexpression inhibits HIF1A,

LOX and ITGA5 in a robust double-negative feedback loop, thus sensitizes cells to chemotherapy. Although we have shown that miR-142-3p is significantly associated with survival specifically in chemotherapy-treated TNBC patients, future studies including miRNA profiling or in situ hybridization in large chemotherapy-treated patient cohorts are warranted to fully uncover the potential of this miRNA as a biomarker and a therapy sensitizer in TNBCs.

In summary, we uncovered a molecular mechanism of chemotherapy resistance in TNBC involving hypoxia-driven LOX expression, which on one hand, leads to increased collagen cross-linking, fibronectin assembly and decreased drug penetration, and on the other hand, increases ITGA5 and FN1 expression, collectively leading to increased FAK/Src signaling, decreased apoptosis and chemoresistance. Furthermore, hypoxia-driven repression of miR-142-3p increases HIF1A, LOX, and ITGA5 expression, leading to further activation of FAK/Src signaling (Fig. 8). Based on these results, we propose that targeting LOX or its downstream FAK or Src or restoring the hypoxia-inhibited miR-142-3p can overcome chemoresistance by effectively blocking FAK/Src signaling (Fig. 8). Overall, our results provide valuable insights into how chemoresistance can be modulated at multiple levels and a pre-clinical support for inhibiting a key ECM-remodeler, LOX in TNBCs to potentiate chemotherapy response.

## Methods

**In vivo experiments**. To develop doxorubicin-resistant xenografts, 6–8-weeks-old female athymic nu/nu mice were injected with $2 \times 10^6$ MDA-MB-231 cells into mammary fat pads (MFP). Mice were randomly allocated into two groups, and vehicle or doxorubicin treatments (5 mg/kg, weekly, i.v.) were started when tumors became palpable. Vehicle-treated tumors were collected and designated as vehicle. Half of the doxorubicin-treated mice were sacrificed when they were responsive to treatment, and tumors were collected and designated as sensitive. The remaining mice continued to receive doxorubicin until their tumors started regrowth. Then, these mice were also sacrificed, and tumors were collected and designated as resistant.

To test the role of LOX or FAK or Src inhibition to overcome chemotherapy resistance in immunocompetent setting, $2 \times 10^5$ 4T1 cells were injected into the MFPs of 6–8 weeks-old Balb/c mice. Mice were treated with vehicle, doxorubicin (2.5 mg/kg, once a week, i.v.), BAPN (100 mg/kg, daily, ip.), PF-562271 (15 mg/kg, daily, oral), Saracatinib (25 mg/kg, daily, oral), or their combinations with doxorubicin. Once the vehicle group reached at 800 mm³, all mice were sacrificed, and tumor weight was measured.

For shLOX induction experiments using the luciferase overexpressing MDA-MB-231.Luc2GFP cells, doxycycline was given to induce shLOX expression at 100 ug/ml in drinking water when the tumors reach at 100 mm³. All mice were sacrificed; tumors were collected and weighed once the vehicle-treated tumors reached at 1500 mm³. For bioluminescence imaging, mice were intraperitoneally injected with 150 mg/kg D-luciferin (Perkin Elmer, MA, USA), and images were acquired with Lumina III In Vivo Imaging System (Perkin Elmer, MA, USA). Analysis was performed with Living image software by measuring photon flux.

For PDX experiments, 2–3 mm³ pieces of frozen PDX tumors were placed into the flank region of NSG mice. When tumors become palpable, mice were distributed into treatment groups. Doxorubicin (2 mg/kg, once a week, i.v.) and BAPN (100 mg/kg, daily, i.p.) was given individually or in combination for 30 days after which mice were sacrificed and tumors were collected for downstream analysis.

Primary tumor growth was monitored by measuring the tumor volume at least twice a week with a caliper after tumors became palpable. Tumor volumes were calculated as length × width²/2. All mice used were of the same age and similar body weight. All animal experiments have been approved by the Animal Ethics Committee of Bilkent University or the Institutional Animal Care and Use

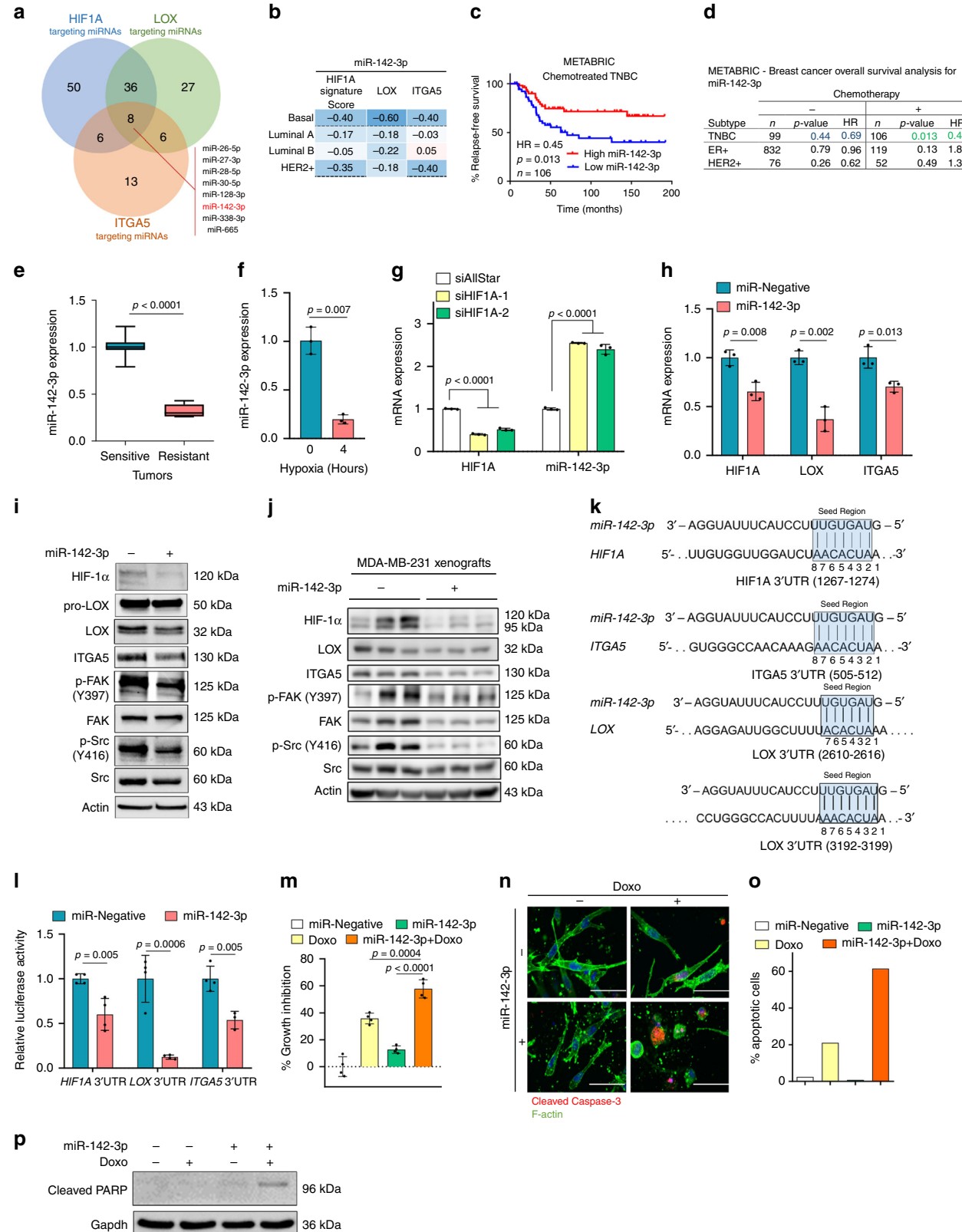

Committee of University of South Carolina. All mice were maintained under a temperature-controlled environment with a 12-h light/dark cycle and received a standard diet and water ad libitum.

**Patient tumor-derived and PDX-derived organoids**. For the generation of patient organoids, breast cancer surgical patients were consented under an IRB approved protocol for the USC-Palmetto Health Biorepository. TNBC organoids were established from a fresh surgical tissue by cutting the tumor into small pieces

and incubating in collagenase A solution with ROCK inhibitor on a shaker at 37 °C for 30 min. The collagenase activity was inhibited by adding FBS, and pipetting was done to ensure the formation of almost a single cell solution. After several washes with PBS, the cell pellet was dissolved in matrigel. Breast organoid media containing ROCK and GSK inhibitors was added after the matrigel solidified[69]. For the PDX organoid, a small, 2–3 mm³ piece of the PDX tumor that was freshly excised from an NSG mouse was minced and separated into single cell suspension as previously described[70]. For drug testing studies, organoids were disrupted to single

**Fig. 7 miR-142-3p regulates HIF1A/LOX/ITGA5 axis to confer chemosensitization in TNBC. a** Venn diagram of combinatorial target prediction analysis. Number of miRNAs targeting *HIF1A* (blue), *LOX* (green) and *ITGA5* (orange) is shown. Eight miRNAs predicted to target all three genes are shown. **b** Heatmap showing Pearson's correlation coefficients between miR-142-3p and *HIF1A* gene signature score, *LOX* and *ITGA5* mRNA expressions in patients from GSE19783. An intense blue color shows a stronger negative correlation. **c** Kaplan–Meier survival curve in chemotherapy-treated TNBC patients ($n = 106$) based on low vs. high (median) miR-142-3p expression. **d** Table summarizing the association of miR-142-3p expression with survival in different breast cancer subtypes with or without chemotherapy. **e, f** qRT-PCR analyses of miR-142-3p expression in doxorubicin-sensitive ($n = 11$) vs. doxorubicin-resistant ($n = 12$) xenografts (**e**) and in MDA-MB-231 cells under hypoxia for 4 h ($n = 3$) (**f**). **g** qRT-PCR of miR-142-3p upon transfection with two different siRNAs targeting HIF1A for 48 h ($n = 3$). **h, i** qRT-PCR ($n = 3$) (**h**) and western blot (**i**) analyses of HIF/LOX/ITGA5 axis upon miR-142-3p transfection. **j**. Western blot analyses of the HIF1A/LOX/ITGA5 axis in MDA-MB-231 xenografts stably expressing miR-142-3p. **k** Graphical representation of miR-142-3p binding sites within the 3'-UTRs of *HIF1A*, *LOX* and *ITGA5*. **l**. Luciferase reporter assay with 3'-UTRs of *HIF1A*, *LOX* or *ITGA5* in MDA-MB-231 cells transfected with miR-Negative or miR-142-3p ($n = 4$ (*HIF1A* and *LOX*), $n = 3–4$ for (*ITGA5*)). **m** Percentage growth inhibition in collagen-embedded MDA-MB-231 cells after transfection with miR-142-3p in the presence or absence of doxorubicin ($n = 4$). **n** Immunofluorescence staining of Cleaved Caspase-3 (red) and F-actin (green) in miR-Negative or miR-142-3p transfected MDA-MB-231 cells in the presence or absence of doxorubicin. **o**. Quantification of Cleaved Caspase-3 positive cells from **n**. **p**. Western blot of cleaved PARP upon miR-142-3p transfection with or without doxorubicin treatment for 72 h. Data represents mean ± SD. Two-sided Student's *t*-test was used to calculate statistical difference between two groups. One-way ANOVA with Dunnett's test was performed to compare mean of combination-treated group with single agent treatments in **m**. Significance for survival analyses was calculated by log-rank (Mantel-Cox) test. HR hazard ratio. Scale bar = 50 μm for **n**. Source data are provided as a Source data file.

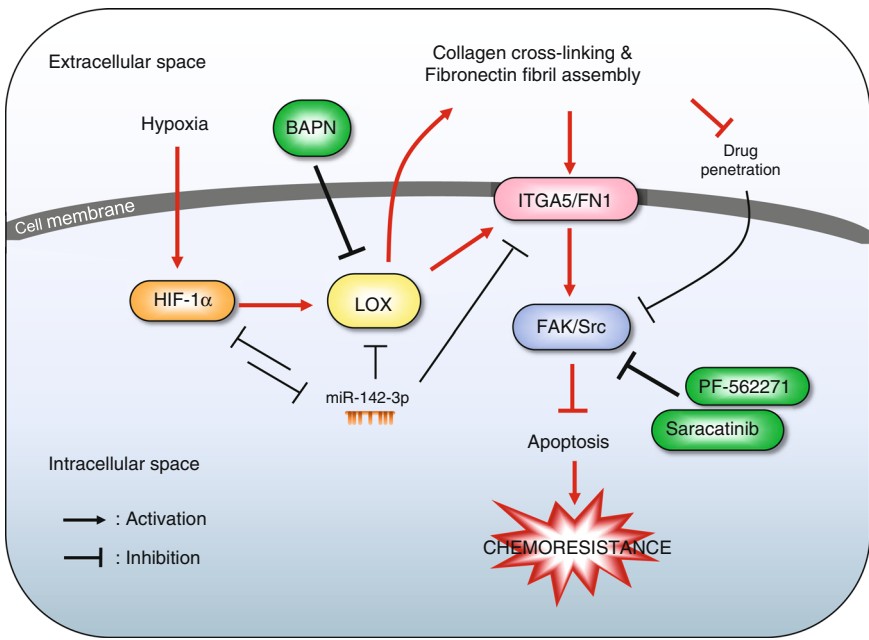

**Fig. 8 Mechanistic summary and targeting approaches for overcoming chemoresistance.** In the hypoxic, chemoresistant TNBC tumor microenvironment, hypoxia induces HIF-1α which then increases the transcription of LOX. LOX, on one hand, increases the expressions of ITGA5 and its ligand, fibronectin in tumor cells and on the other hand, it induces collagen cross-linking and fibronectin fibril assembly leading to reduced drug penetration into tumor cells. In meantime, hypoxia-mediated downregulation of miR-142-3p, which directly targets HIF1A, LOX and ITGA5, leads to further activation of the HIF1A/LOX/ITGA5/FN1 axis. Overall, this culminates in the activation of FAK/Src signaling, blockage of drug-induced apoptosis and chemoresistance in TNBCs. Therefore, using inhibitors targeting LOX (e.g. BAPN) or its downstream FAK (e.g. PF-562271) or Src (Saracatinib) could overcome chemoresistance in TNBCs.

cells by digesting at 37 °C for 30 min with TrypLE (A1217701, Gibco, NY, USA) in the presence of 10uM Rock inhibitor (s1049, Selleckchem, TX, USA). Organoids were plated into wells of 96-well plate (20,000 cells/well) on a collagen-coated surface with media containing 2% matrigel (356252, Corning, NY, USA). Drugs were added 72 h after plating (final concentration of doxorubicin at 40 nM and BAPN at 25 mM). Cells were grown in the presence of drug(s) or vehicle for 9 days, and the diameter of organoids were measured.

**Whole-transcriptome sequencing (RNA-seq) and data analyses.** rRNA-depleted stranded libraries for each condition (four biological replicates for each of vehicle, doxorubicin-sensitive and doxorubicin-resistant tumors) were generated and multiplexed. Paired-end 100 bp sequencing was performed using the Illumina HiSeq 2000 platform at McGill University and Genome Quebec Innovation Centre. In all, 60–70 million sequencing reads were obtained for each sample. Raw sequence reads were aligned to the UCSC human reference genome (hg19) using TopHat. with default parameters. To count the mapped reads, HTSeq was used with the reference genome annotation (USCS, hg19). In order to

determine differentially expressed genes between doxorubicin-sensitive and doxorubicin-resistant groups, Bioconductor package, edgeR, was utilized.

**Patient data analyses.** To analyze the effects of LOX expression on the survival of chemotherapy-treated TNBCs, we performed IHC staining of LOX in primary tumor samples from 77 TNBC patients that were diagnosed with breast cancer between 2006 and 2015 at Hacettepe University School of Medicine, Ankara, Turkey and treated with chemotherapy (35% adjuvant anthracycline-based therapy, 43% anthracyclines in combination with taxanes and 22% other chemotherapy agents). The study was approved by the Non-Interventional Clinical Research Ethics Committee of Hacettepe University (approval no: 2020/02-40). To generate DoxoR-GS scores in patients from METABRIC, GSE31519 and GSE58812, we used the top 441 differentially (up or down-regulated) expressed mRNAs between doxorubicin-sensitive vs. -resistant xenografts with a fold change cut-off of 1.75 and *p*-value cut-off of 0.05 (Supplementary Data 1). First, expression of these 441 genes in patients were converted into *z*-scores. Then, the sum of *z*-scores of the downregulated genes in the DoxoR-GS was subtracted from the sum of *z*-scores of

the upregulated genes for each patient[35] using the SPSS Statistics software. Chemotherapy-treated patient subgroup from the KM Plotter database (release version 2017) were created by selecting the adjuvant chemotherapy-treated patients and excluding the endocrine-treated ones. In GSE19783 dataset, patients are stratified based on the PAM50 subtyping (basal patients were selected) and ERα, PR and HER2 expression.

**Statistics and reproducibility**. The results are represented as mean ± standard deviation (SD) or mean ± standard error of the mean (SEM), as indicated in the figure legends. All statistical analyses were performed in GraphPad Prism Software using two-sided Student's t-test for comparisons between two groups, one-way ANOVA followed by Dunnett multiple comparison test for comparing combination treatment group with single agent-treated groups, and two-way ANOVA for comparing tumor volume change over time among different treatment groups. Survival curves were generated based on median separation using Kaplan–Meier method, and significance between groups was calculated by Log-rank test. For correlation analysis, Pearson correlation coefficients were calculated. Experiments were repeated two to three times independently with similar results.

Methods for cell culturing, cell-based assays, transient and stable transfections, qRT-PCR and Western blot analysis, immunofluorescence, immunohistochemical and Picrosirius red staining, Annexin V/DAPI staining, hypoxia, cloning, dual luciferase, LOX activity, collagen and deoxycholate lysis assays, other animal experiments, measuring doxorubicin penetration in vitro and in vivo, and patient data and pathway analyses were provided in Supplementary Methods. The list of primers for qRT-PCR, antibodies for Western blotting, IF and IHC as well as the primer sequences for 3′-UTR cloning are provided in Supplementary Table 4.

**Reporting summary**. Further information on research design is available in the Nature Research Reporting Summary linked to this article.

## Data availability

The RNA-Seq data has been uploaded to the Sequence Read Archive (SRA) of NCBI with the submission ID: SUB6918779 (https://www.ncbi.nlm.nih.gov/bioproject/PRJNA607780). Data presented on Figs. 1d, f, 2b and Supplementary Figs. 2b, c, 4a, b were generated by analyzing the data available under the accession number GSE58812 from GEO depository (https://www.ncbi.nlm.nih.gov/geo/). Data presented on Figs. 1h, i, k, l, 2o and Supplementary Figs. 1d, e, 2a, d were generated by analyzing the KM Plotter database (http://kmplot.com/analysis/). Data presented on Fig. 7c, d and Supplementary Figs. 1a, 12d were generated by analyzing the METABRIC data from EMBL European Genome–Phenome Archive (http://www.ebi.ac.uk/ega/) with an accession number EGAS00000000122. GSEA gene sets were downloaded from the GSEA MSigDB Collections website: https://www.gsea-msigdb.org/gsea/msigdb/collections.jsp. Data presented on Fig. 7b and Supplementary Fig. 12a, b, c were generated by analyzing the data available under the accession number GSE19783 from NCBI. Data presented on Supplementary Figs. 1b, 5a were generated by analyzing the data available under the accession number GSE31519 from NCBI. Data presented on Supplementary Fig. 5b–e were generated by analyzing the data available under the accession number GSE25066 from NCBI. Data presented on Fig. 2h was generated by analyzing the data available under the accession numbers GSE16446, GSE19783, GSE21653, GSE22219, GSE22226, GSE25066, GSE58644 from NCBI and under METABRIC datasets and The Cancer Genome Atlas (TCGA) data. The source data underlying Figs. 1a–h, j, k; 2a–e, g–o, q; 3a–g, i, k–q; 4a–e, g–j, l, n–p, r; 5a, c–e, g, i, k–l; 6a, c, e–f, h, i, k, m, o, q, s; 7b, c, e–j, l, m, p and Supplementary Figs. 1a–c; 2a–c; 3a–b; 4c, d; 5c–e; 6b, f; 7a–d; 8a, c; 9a, c; 10a–c, e, f; 11b–f; 12e are provided as a Source Data file. All the other data supporting the findings of this study are available within the article and its Supplementary Information files and from the corresponding author upon reasonable request. A reporting summary for this article is available as a Supplementary Information file.

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

## Acknowledgements

We thank Madeleine Arseneault and Ozlem Sener Sahin for technical and organizational support, and Jitao David Zhang for his help with the initial analysis of RNA-Seq data. We also thank the Center for Targeted Therapeutics (CTT) Microscopy and Flow Cytometry Core (Dr. Chang-uk Lim) of the University of South Carolina, for assistance with flow cytometry. This work was supported by National Institutes of Health Grant 2P20GM109091-06 (O.S.), and in part by European Commission FP7 Marie Curie Career Integration Grant PCIG14-GA-2013-631149 (O.S.), European Molecular Biology Organization (EMBO) Installation Grant 2791 (O.S.), the American Cancer Society Institutional Research Grant IRG-17-179-04 (O.S.), the American Cancer Society Research Scholar Grant RSG-19-194-01-CSM (O.S.), Susan G. Komen Interdisciplinary Graduate Training to Eliminate Cancer Disparities (IGniTE-CD) GTDR17500160 (Ozge S.), the scholarship from Higher Education Commission of Pakistan (U.R.), and research scholarship of the Fonds de recherche du Québec—Santé (FRQS) (Y.R.).

## Author contributions

Ozge S., A.K., and U.R. designed and performed experiments, acquired and analyzed data, interpreted data, and prepared the paper; P.G.E. and O.A. contributed to in vivo experiments, IHC staining and data collection; C.E.B. generated TNBC patient organoids; V.S. contributed to confocal microscopy experiments and data analyses; S.A.A. performed cloning and luciferase reporter experiments; U.M.T. contributed to RNA-seq analyses; G.A. contributed to in vivo experiments; H.T.D., M.D., performed the IHC staining of xenografts and contributed to data interpretation; P.J. performed RNA-sequencing; A.I., F.G., K.K., A.A., and A.U. performed the IHC staining of TNBC patient tissues and contributed to data interpretation; O.D. and S.A. provided clinical information of TNBC patients and contributed to data analyses; P.J.B. contributed to TNBC tumor organoid experiments and critically read and edited the manuscript; Y.R. performed RNA-sequencing, and critically read and edited the manuscript; O.S. designed the study, oversaw experiments and data analyses, and prepared the paper. All authors reviewed and commented on the paper.

## Competing interests

The authors declare no competing interests.
