## [Peer Review File · Nature Communications]

Reviewers' comments:

Reviewer #1 (Remarks to the Author):

In their manuscript, Saatci and colleagues show a novel mechanism of doxorubicin resistance in triple negative breast cancer (TNBC) that involves the activation of a HIF1A/miR-142-3p/LOX/ITGA5/FN1 signaling axis. The authors show that activation of this signaling axis leads to chemoresistance in several models for TNBC, including experiments performed with tumor cell lines in vitro and in vivo, an organoid model from a treatment-naïve TNBC patient and publicly-available breast cancer datasets. These findings are potentially very interesting, but if the authors were to deal with the following comments, the manuscript could be significantly strengthened.

Specific comments:

1. The authors should include the tumor growth curves of the control-treated, doxorubicin-sensitive and -resistant tumors in figure 1. This is necessary as the authors should show the effect of doxorubicin on tumor growth and the timescale in which these tumors developed resistance to the drug.
2. The authors should provide enlarged images/images with a greater magnification of the immunohistochemistry studies to highlight the exact location of the protein of interest in the tumor cells. The current images do not provide sufficient detail.
3. In figure 2c the authors show that the growth inhibition of MDA-MB-231 cells is 50% in normoxia and 40% in hypoxia. Based on these results, the authors should not emphasize that the cells are resistant to doxorubicin. Although the cells seem to be slightly less sensitive to doxorubicin, the growth inhibition shows that the cells are still sensitive to the drug.
4. The authors show in figure 3b that p-Src and p-FAK are less activated when the cells are embedded in collagen compared to "no collagen". Is this correct? The samples be run on the same gel to make a direct comparison. This is important because if these phosphorylated proteins are less activated (or only modestly activated) when the cells are embedded in collagen, this might indicate that the cells are less dependent on this type of signalling under these conditions. This could therefore be a confounding factor when testing the effect of doxorubicin on these phosphorylated proteins.
5. Based on the immunofluorescence images shown in figure 3f, it is surprising that the cells treated with BAPN and doxorubicin have the highest percentage of apoptotic cells. The cells appear less positive for cleaved caspase-3 as compared to the other three panels. The authors should elaborate on this and describe the exact method (in the Material and Methods) used to quantify the positive cells resulting in such a profound effect as shown in the quantification panel. The data presented in figures 3c, 3f and 6m would be more quantitative and informative if the cells were stained for Annexin V and Propidium Iodide (PI) and subsequently analysed by flow cytometry.
6. It is reasonable for the authors to investigate the effect of doxorubicin-related autofluorescence in their experiments (supplementary figure 6), but to that point, they should have investigated this effect in the cleaved caspase-3 staining conditions used in the experiments rather than the rather different, and probably much stronger conditions (phalloidin) they present.
7. The scalebars in the panels of figure 3h are missing. Based on the presented images it seems that the "only ECM" panels were acquired using a different magnification to the 'Vehicle' and 'BAPN' panels. Furthermore, the authors should show the position of the cells in these panels to highlight the contact (or absence of contact) of the cancer cells with fibronectin/collagen in the vehicle and BAPN-treated conditions.
8. The left LOX 3'UTR in figure 6j has a shorter highlighted seed region (6 nt) compared to the other 3 (7 nt). Is this correct?

Reviewer #2 (Remarks to the Author):

Saatci and colleagues report that de-regulation of ECM signaling involving the miR-142-3p/HIF1A/LOX/ITGA5/FN1/FAK/Src axis drives chemotherapy resistance in triple negative breast cancer (TNBC). The authors validate their in vitro and in vivo experimental models in human breast cancer datasets and propose targeting LOX as a therapeutic strategy to treat chemo-resistant TNBC.

Overall, the results are interesting, however, the study critically lacks controls and is need of more clinically relevant data/discussion to make it more suitable for publication.

Comments:

1. The vast majority of observations in vitro are driven by the use of only one cell line (MDA-MB-231). The study's conclusions would be benefit substantially from a more comprehensive characterization across additional TNBC cell line models, including negative control lines (non-TNBC).
2. The use of patient-derived xenograft models is lacking and would significantly increase the clinical impact and relevance of the study.
3. Fig 1f, i: The effect of ITGA5 and FN1 together should be examine as prognostic factors instead of separately and also confirmed in additional independent human datasets.
4. LOX expression is missing across the experimental models tested in Figs. 3-5. This precludes subsequent conclusions concerning the causal role of LOX in driving downstream effects.
5. Fig 3i: Biochemical data is missing to show FAK and SRC inhibitors reduce phospho-levels as was shown in previous panels Fig 3b, e. Clinical implications of the study would be greatly improved if the combinatorial effect of these drugs were validated in animal models.
6. Fig 6c: How was high and low miR-142-3p defined for the KM analysis? The authors should also confirm their findings in additional datasets.
7. The consequences of miR-142-3p on FAK and SRC should be examined in vitro and in vivo.
8. LOX has been well-reported to have roles in metastasis, the process that drives lethality in breast cancer. The authors should strongly consider experiments/discussion on the impact of targeting LOX and/or downstream components such as FAK/SRC on metastatic biology.
9. It remains unclear the proportion of chemo-resistant patients that may benefit from LOX therapy. This should be included in the discussion.

Reviewer #3 (Remarks to the Author):

In this manuscript the authors report that the ECM remodelling protein LOX is a key player in resistance of triple negative breast cancer (TNBC) to standard chemotherapy through the LOX/ITGA5/FN1/FAK/Src axis. They base the initial hypothesis in a single cell line MDA-MB-231 TNBC cells xenografted in mice, which is poorly representative of TNBC. Among the upregulated pathways in MDA-MB-231 doxorubicin resistant tumors they find integrin signaling leading to increased cell survival. The authors propose that resistant tumors become so by activating hypoxia signaling, which in turn upregulates LOX, ITGA5 and FN1 genes; Saatci et al. They propose that LOX inhibition overcomes resistance both in vitro and in vivo. They finally propose miR-142-3p as an upstream regulator of the hypoxia-LOX-integrin signaling axis responsible for chemoresistance in TNBC.

The manuscript is not easy to follow and seems a patchwork, where several disconnected results

have been pushed together; authors keep jumping from one hypothesis to the next with poor links between them. The MS multiple hypotheses are not fully sustained by the data shown herein and multiple key points would be required in order to functionally demonstrate the involvement of the miR-142-3p-HIF1 α -ITG5A-FN1 in TNBC chemoresistance and to strengthen the links between them.

1. It is unclear why the authors compare doxorubicin resistant tumors and “residual disease” to generate the DoxoR-GS. Cells that survive short term doxorubicin treatment are also resistant. How many independent tumors were used for RNAseq. MDA-MB-231 is not representative of clinical TNBC. A welcome strategy would be to include additional cell lines and Patient derived-xenograft (PDX), which would improve the reliability of the results.

2. In Fig 1j authors claim a significant association between high FN1 and poorer RFS in chemotherapy-treated basal patients, but $p = 0.05$, indicating no significance. Moreover, Are the chemotherapy-treated TNBC patient datasets in which the DoxoR-GS score was analyzed from patients with analogous clinical management? This might be relevant since the tumors in mice were homogenously generated using doxorubicin and not with other chemotherapies such as platinum, taxanes, or olaparib. It would be relevant to know whether the proposed LOX-mediated mechanism is involved in chemoresistance beyond doxorubicin.

3. Have the authors checked the levels of LOX, ITGA5 and FN1 in the vehicle tumors generated in mice from 231 cells, to demonstrate that these genes are only overexpressed in chemoresistant tumors vs sensitive ones?

4. In Fig. 2g the authors should check FAK and Src total levels in order to conclude that their phosphorylation levels are upregulated. The same is true for the blots in Fig. 2j, 3b and 3e. Is this LOX upregulation also seen in 231 cells upon doxorubicin treatment? Have the authors attempted to overexpress LOX in 231 cells and check for their sensitivity towards doxorubicin? That would help sustain LOX claimed role.

5. In figure 2j phosphorylation of FAK and Src is downregulated when LOX is inhibited, however, this downregulation is not observed in the presence of the LOX inhibitor BAPN in Figure 3e. Authors should explain these discrepancies. Moreover, LOX silencing as shown in Fig. 2j (32kDa) is too weak. A stable or stronger silencing is required to conclude it has a downstream impact.

6. LOX activity is not easy to measure, the data shown in the graph 2h lacks negative controls (such as adding BAPN inhibitor). How is LOX activity in resistant tumors? That would be more interesting than analyzing it in cell culture.

7. BAPN is not a selective inhibitor of LOX since it also has activity against other LOX family members. The authors have not checked the levels of those proteins, particularly LOXL2, in 231 cells, where it has been shown to be expressed and have a prominent role (Moreno-Bueno et al., 2011). LOX levels should also be shown.

8. What is the effect on LOX and HIF1 α expression in cells cultured with doxorubicin in the presence of the absence of collagen? (Figure 3b).

9. The graph on the right of images in Fig. 3g is not properly labelled and does not seem to account for the differences displayed by the images. Besides, these immunofluorescence images cannot account as a formal method to analyse collagen and FN deposition. The same is true for the experiments shown in Fig. 3h. The results displayed in Fig. 3g and 3h are not adequate to back up the authors' conclusion regarding “a prominent decrease in collagen cross-linking” or “significant reduction in fibronectin deposition” (page 15). These are few images that might not be representative. Additional images should be acquired per experiment with replicates, and intensity quantified using adequate software (i.e. as performed in Qureshi et al, Biol Open 2017) and an alternative method is required. For Figure 3h, statistical analysis is required to state that there is a strong co-localization of type I collagen and fibronectin fibrils?

10. Moreover, in Fig. 3h there are no clear differences in vehicle vs BAPN in the images displayed. In this line in page 10, second paragraph: with the reported results, the authors cannot state "...demonstrated the reversal of collagen cross-linking upon LOX inhibition...". For that, further biochemical experiments are required.

11. In Figure 3i authors show the percentage of growth inhibition of doxorubicin in the presence of FAK and Src inhibitors, but what is the percentage of apoptotic cells? This needs to be clarified since authors conclude: "LOX-mediated collagen cross-linking and the subsequent increase in fibronectin fibril assembly contribute to chemotherapy resistance via activation of FAK/Src signaling and inhibition of apoptosis in chemotherapy-treated TNBCs".

12. In 4T1 cells the authors should check the levels of LOX and LOX family members, at least LOXL2, since the effects observed upon BAPN treatment could be due to LOXL2 inhibition. It might be possible that LOX is not expressed at all in these tumors and the LOX activity shown in Fig. 4h is not due to LOX itself.

13. Figure 4. How the authors explain the combinatory effect of doxorubicin and BAPN in the chemoresistant TNBC model when the inhibition of LOX activity is similar with BAPN and doxorubicin plus BAPN? Another alternative pathway has to be activated in order to explain the combinatory mechanism.

14. The results shown in Fig. 5 are not sound unless LOX protein levels are shown upon shRNA interference. What is the effect of combinatory treatment on pFAK? And the percentage of apoptotic cells?

15. Figure 6b. Is there a p-value for the strong inverse correlation between the miRNAs miR-142-3p and miR-128-3p and HIF1alpha, LOX and ITGA5 signatures? Figure 6i. What is the effect of miR-142-3p on pFAK and pSrc?

16. The authors claim that LOX might be triggering ITG5A and FN1 expression without formally proving it and at the same time they propose that is miR-142-3p the one that regulates their expression. There are not enough data in the manuscript to sustain these links.

17. In the Discussion, the authors claim that their results "underpin the intense communication between resistant tumor cells and surrounding ECM" (page 15) which is an overstatement given the fact that their results derived mostly from cells in vitro cultured and they do not show any ECM modification within the tumors.

In summary, although Saatci and co-workers propose an interesting mechanism, they do not demonstrate that LOX is indeed responsible for the TNBC chemoresistance, and the same occurs with most claims. Furthermore, their conclusions are not clear since they mention several mechanisms such as hypoxia, enhanced ECM stiffness or LOX upregulation and subsequent downstream signaling. None of this is formally proven within the tumors.

MINOR POINTS

Revise Figure legends (ie. In Fig. 3g the graph on the right is not mentioned within the text nor the axes are not properly labelled)

In Fig. 4f the images displayed suggest an increase in size for Doxo treated tumors vs Ctrl.

The journal titles in the reference should be corrected.

Doxorubicin resistant xenografts" should be "doxorubicin-resistant xenografts".

ANSWERS TO REVIEWERS' COMMENTS

We thank the reviewers for their insightful comments, and we greatly appreciate that they found the original manuscript interesting. We have performed extensive additional experimental work and data analyses to validate our proposed mechanism of chemoresistance in TNBCs that involve *hypoxic upregulation of LOX leading to both enhanced collagen cross-linking/reduced drug penetration and increased ITGA5/FN1 expression, culminating in activation of downstream FAK/Src signaling and inhibition of apoptosis*. We believe that addressing the reviewers' comments with additional data has significantly improved the quality of our work, and we hope the reviewers now find our improved manuscript suitable for publication in *Nature Communications*. In response to the reviewers' comments, we provided the following data/analyses to further support our initial findings. Briefly:

- We included several other breast cancer cell lines (both TNBC and non-TNBC) and other chemotherapy agents, which are used in clinics to treat TNBCs, to further validate LOX inhibition-mediated chemosensitization.
- We performed *in vivo* experiments with well-characterized, chemotherapy resistant TNBC patient-derived xenografts (PDXs) to show LOX inhibition-mediated chemosensitization in a more clinically-relevant setting.
- We performed new *in vivo* experiments with syngeneic, chemoresistant TNBC models to validate the roles of FAK and Src inhibition in chemosensitization and obtained results that support our proposed mechanism *in vivo*.
- We performed new chemosensitization experiments with 3D PDX-derived and patient-derived organoids to increase clinical relevance of our findings.
- We stained the LOX protein in tissues of 77 TNBC patients treated with chemotherapy and showed that higher LOX protein levels were significantly associated with worse patient survival, increasing clinical relevance of our findings.
- We included new biochemical and microscopy data with quantifications to properly assess the changes in collagen cross-linking, fibronectin assembly and drug penetration both in *in vitro* and *in vivo* settings, further supporting our proposed mechanism.
- We examined LOX expression/knockdown/activity and the expression of other LOX family members, e.g. LOXL2 in *in vitro* and *in vivo* settings to further prove that our proposed mechanism is specifically driven by LOX.
- We performed new xenograft experiments to show the effect of miR-142-3p on HIF1 α /LOX/ITG5A and downstream FAK/Src signaling *in vivo*.

In order to accommodate all these new supporting evidences, we have included a new main figure (Figure 5), several new figure panels, five new supplementary figures and one supplementary table, and incorporated these changes in results, methods and discussion sections (marked in red).

Furthermore, we provided a detailed point-by-point response to each reviewer's questions below.

Reviewer #1 (Remarks to the Author)

In their manuscript, Saatci and colleagues show a novel mechanism of doxorubicin resistance in triple negative breast cancer (TNBC) that involves the activation of a HIF1A/miR-142-3p/LOX/ITGA5/FN1 signaling axis. The authors show that activation of this signaling axis leads to chemoresistance in several models for TNBC, including experiments performed with tumor cell lines *in vitro* and *in vivo*, an organoid model from a treatment-naïve TNBC patient and publicly-available breast cancer datasets.

These findings are potentially very interesting, but if the authors were to deal with the following comments, the manuscript could be significantly strengthened.

Response: We thank the reviewer for evaluating our manuscript, finding our work very interesting, and the comments she/he made to strengthen our manuscript. Below, we provided point-by-point responses to reviewer's specific comments.

Specific comments:

1. The authors should include the tumor growth curves of the control-treated, doxorubicin-sensitive and -resistant tumors in figure 1. This is necessary as the authors should show the effect of doxorubicin on tumor growth and the timescale in which these tumors developed resistance to the drug.

Response: As the reviewer suggested, we added the tumor growth curves in mice from vehicle-treated, doxorubicin-sensitive and -resistant groups to **Fig. 1b** of the revised manuscript. We also provided the Waterfall plot showing the tumor volume fold change over time for each individual tumor (**Fig. 1c**). In this figure, the tumors labelled with asterisks are the ones used for RNA-Seq; n=4 for vehicle, sensitive and resistant groups. Accordingly, the growth rate of resistant tumors is similar to the growth rate of vehicle tumors after the development of resistance, whereas sensitive tumors exhibit tumor shrinkage over time and have a negative tumor volume fold change (**Fig. 1c**). This is now described in Page 6, Lines 129-131 in the revised manuscript. Furthermore, RNA-Seq data has been uploaded to the Sequence Read Archive (SRA) of NCBI with the submission ID: SUB6918779 in accordance with the *Nature Communications* guidelines.

2. The authors should provide enlarged images/images with a greater magnification of the immunohistochemistry studies to highlight the exact location of the protein of interest in the tumor cells. The current images do not provide sufficient detail.

Response: In the revised version of the manuscript, we provided enlarged images of the immunohistochemistry (IHC) staining and mentioned the exact locations of the proteins of interest in the corresponding figure legends. We also provided new images for LOX and ITGA5 in sensitive vs. resistant tumors in new **Fig. 1g** and **Fig. 2e** that better demonstrates the staining patterns of the antibodies. Accordingly, we observed membranous and cytoplasmic ITGA5 staining (**Fig. 1g**); mild to moderate cytoplasmic FN1 staining (**Fig. 1j**); predominantly membranous and mild cytoplasmic CA9 staining (**Fig. 2a**); and strong cytoplasmic and weak nuclear LOX staining (**Fig. 2e**).

3. In figure 2c the authors show that the growth inhibition of MDA-MB-231 cells is 50% in normoxia and 40% in hypoxia. Based on these results, the authors should not emphasize that the cells are resistant to doxorubicin. Although the cells seem to be slightly less sensitive to doxorubicin, the growth inhibition shows that the cells are still sensitive to the drug.

Response: We agree with the reviewer that it is difficult to conclude that cells are resistant to doxorubicin based on the growth inhibition plot we provided in the initial version of the manuscript. In order to show the effect of hypoxia on doxorubicin response more clearly, we repeated this experiment by changing the hypoxic conditions from 5% oxygen to 1% oxygen as previously reported¹. As shown in **Fig. 2c** of the revised manuscript, reduction in doxorubicin response is more pronounced when cells are cultured under 1% oxygen and is now more convincing. In addition, we examined the changes in doxorubicin response under hypoxic conditions using another TNBC cell line, MDA-MB-157 to rule out any cell line-specific effect. We again observed reduced doxorubicin response at a similar extent to MDA-MB-231 in these cells (**Fig. 2d**). The increase in HIF-1 α protein levels, upregulation of LOX and ITGA5, as well as the

activation of downstream FAK/Src signaling have also been validated in MDA-MB-157 cells (**Supplementary Fig. 4c**) and are similar to our initial results with MDA-MB-231 cells. These results strengthen our hypothesis that hypoxia induces LOX and ITGA5 signaling to confer chemoresistance. This is now described in Page 8, Lines 183-185 and Page 9, Lines 210-212 in the revised manuscript.

4. The authors show in figure 3b that p-Src and p-FAK are less activated when the cells are embedded in collagen compared to “no collagen”. Is this correct? The samples be run on the same gel to make a direct comparison. This is important because if these phosphorylated proteins are less activated (or only modestly activated) when the cells are embedded in collagen, this might indicate that the cells are less dependent on this type of signalling under these conditions. This could therefore be a confounding factor when testing the effect of doxorubicin on these phosphorylated proteins.

Response: We thank the reviewer for this insightful comment. In order to be able to compare the signaling in cells grown in uncoated vs. collagen-embedded conditions, we have re-run the gels with all samples together. As shown in new **Fig. 3c** of the revised manuscript, phosphorylation of FAK and Src are more pronounced in collagen-embedded cells, suggesting that FAK/Src signaling is more activated when the LOX substrate, collagen I is in the environment. Importantly, doxorubicin caused a decrease in FAK/Src signaling activity in cells grown in uncoated conditions whereas phosphorylation of FAK/Src was mostly preserved in collagen-embedded, doxorubicin-treated cells, suggesting that cells are more dependent on FAK/Src signaling in collagen, and doxorubicin is less effective in collagen-embedded cells. This is now provided in Pages 11, Lines 245-246 in the revised manuscript.

5. Based on the immunofluorescence images shown in figure 3f, it is surprising that the cells treated with BAPN and doxorubicin have the highest percentage of apoptotic cells. The cells appear less positive for cleaved caspase-3 as compared to the other three panels. The authors should elaborate on this and describe the exact method (in the Material and Methods) used to quantify the positive cells resulting in such a profound effect as shown in the quantification panel. The data presented in figures 3c, 3f and 6m would be more quantitative and informative if the cells were stained for Annexin V and Propidium Iodide (PI) and subsequently analysed by flow cytometry.

Response: We apologize for not properly mentioning the colors of the individual proteins that led to confusion. This figure is now **Supplementary Fig. 6e** in the revised manuscript. Cleaved Caspase-3 was stained with an Alexa Fluor 647-labelled secondary antibody (red), whereas phalloidin was stained with an Alexa Fluor 488-labelled secondary antibody (green). We also provided the quantifications of the percentage of Cleaved Caspase-3 positive cells; i.e., cells that gave a red signal at the bottom of the images representing apoptotic cells (**Supplementary Fig. 6d**). To quantify cleaved caspase-3 positive cells, we set the intensity of the red fluorescence from vehicle-treated cells to zero and calculated the number of cells that gave red signal in each group. We added a detailed explanation of how the percentage of Cleaved Caspase 3-positive cells was calculated to the Supplementary Materials and Methods section of the revised manuscript (Page 6, Lines 129-132).

To provide a more quantitative data for the apoptosis induction upon doxorubicin treatment in 1.) uncoated vs. collagen-embedded cells (new **Fig. 3b** and **Supplementary Fig. 6c**) and 2.) doxorubicin+BAPN-treated cells (new **Fig. 3p** and **Supplementary Fig. 6g**), we performed Annexin V/DAPI staining and validated apoptosis results initially obtained with Cleaved Caspase-3 IF staining. We did not use Propidium Iodide (PI), which is the commonly used viability marker, in conjunction with Annexin V since the excitation/emission wavelengths for PI overlaps with that of doxorubicin. Instead, we used DAPI. Furthermore, to show the effect of miR-142-3p in combination with doxorubicin on apoptosis in collagen I-embedded MDA-MB-231 cells, we performed Western blot for cleaved PARP and

observed its induction in the presence of miR-142-3p and doxorubicin (new **Fig. 7p**), validating Cleaved Caspase-3 IF staining (new **Fig. 7n, o**) in the revised manuscript. This is now described in Page 11, Line 245, Page 13, Line 298-299 and Page 17, Line 397-398 in the revised manuscript.

6. It is reasonable for the authors to investigate the effect of doxorubicin-related autofluorescence in their experiments (supplementary figure 6), but to that point, they should have investigated this effect in the cleaved caspase-3 staining conditions used in the experiments rather than the rather different, and probably much stronger conditions (phalloidin) they present.

Response: In all the apoptosis-related immunofluorescence staining, the red color represents Cleaved Caspase-3, while green color represents phalloidin. We detected doxorubicin autofluorescence mainly in the green channel (the channel used for phalloidin), but not at the laser power and gain used for phalloidin imaging. For the cleaved caspase-3 channel, the autofluorescence was even lower, and we did not detect signal even when the laser power was increased to 15 and gain to 700. This is in line with the expected autofluorescence of doxorubicin³. We moved this part to Supplementary Methods (Page 6, Lines 132-136). We also added the colors of the individual proteins to all the figure legends and Materials and Methods section of the revised manuscript.

7. The scalebars in the panels of figure 3h are missing. Based on the presented images it seems that the “only ECM” panels were acquired using a different magnification to the ‘Vehicle’ and ‘BAPN’ panels. Furthermore, the authors should show the position of the cells in these panels to highlight the contact (or absence of contact) of the cancer cells with fibronectin/collagen in the vehicle and BAPN-treated conditions.

Response: We replaced the images with new images having scale bars (50 μm) and that also show the positioning of cancer cells within the ECM (new **Fig. 3j**). In the revised version, we also quantified the fibronectin and collagen intensities within the ECM that is in contact with vehicle- vs. BAPN-treated cancer cells. We observed reduced intensity for both fibers in the ECM for BAPN-treated cells (new **Fig. 3k**). All images were taken at the same magnification. The staining results were further biochemically validated by quantifying the hydroxyproline residues in the cross-linked collagen (new **Fig. 3l**) as well as the assembled fibronectin by the deoxycholate lysis method (new **Fig. 3m**). This is now described in Page 12, Lines 274-290 in the revised manuscript.

To show cancer cells in contact with ECM, we used DAPI staining as the cell marker and quantified the contact between cells and the surrounding ECM using the ImageJ software⁴. To this end, we constructed the 3D images of both vehicle- and BAPN-treated cells (new **Supplementary Fig. 8b**) and measured the distance between DAPI and fibronectin fibers as detailed in Supplementary Methods section (Page 6-7, Lines 141-144). As shown in new **Supplementary Fig. 8c**, almost 80% of the cells were in close contact with fibronectin in both vehicle- and BAPN-treated conditions. These results suggest that cells are still in contact with ECM under LOX inhibition even though the collagen cross-linking and fibronectin deposition are disrupted. This is now provided in Page 12, Lines 283-286 in the revised manuscript.

8. The left LOX 3'UTR in figure 6j has a shorter highlighted seed region (6 nt) compared to the other 3 (7 nt). Is this correct?

Response: Yes, the first binding site on LOX 3'-UTR for miR-142-3p is a “7mer-1A” seed region, which is defined as 6 matching nucleotides followed by an ‘A’. The other seed regions are “8mer” seed regions and are defined as 7 matching nucleotides followed by an ‘A’⁵. We had only highlighted the matching

nucleotides in the initial manuscript; however, in the revised version, we numbered the nucleotides in the seed regions, as well as the following 'A's after the matching sequences (new **Fig. 7k**).

Reviewer #2 (Remarks to the Author):

Saatci and colleagues report that de-regulation of ECM signaling involving the miR-142-3p/HIF1A/LOX/ITGA5/FN1/FAK/Src axis drives chemotherapy resistance in triple negative breast cancer (TNBC). The authors validate their *in vitro* and *in vivo* experimental models in human breast cancer datasets and propose targeting LOX as a therapeutic strategy to treat chemo-resistant TNBC.

Overall, the results are interesting, however, the study critically lacks controls and is need of more clinically relevant data/discussion to make it more suitable for publication.

Response: We thank the reviewer for evaluating our manuscript, finding our work interesting, and for the comments she/he made to further strengthen our manuscript. We provided necessary controls, including the expressions of LOX and other LOX family members in *in vitro* and *in vivo* models, validation of knockdowns, overexpressions and inhibitors and testing the effect of LOX inhibition in other breast cancer cell lines (TNBC vs. non-TNBC). Furthermore, we provided more clinically relevant data, including more patient data (response and prognosis), IHC staining of LOX protein in tumors from TNBC patients, *in vivo* experiments with patient-derived xenografts (PDX) as well as patient- and PDX-derived 3D tumor organoids. We also tested the effect of FAK and Src inhibitors on chemotherapy response in syngeneic TNBC models *in vivo*. We further modified the Results and Discussion sections accordingly. Below, we provided point-by-point responses to reviewer's specific comments.

Comments:

1. The vast majority of observations *in vitro* are driven by the use of only one cell line (MDA-MB-231). The study's conclusions would benefit substantially from a more comprehensive characterization across additional TNBC cell line models, including negative control lines (non-TNBC).

Response: We thank the reviewer for this insightful comment. In order to demonstrate that our results are not cell line-specific, we first analyzed LOX expression in a panel of breast cancer cell lines from different molecular subtypes. As shown in new **Supplementary Fig. 7a**, LOX is higher in TNBC cell lines in general compared to HER2-positive and ER+ cell lines. Among different TNBC cell lines, we selected MDA-MB-157 and MDA-MB-436 cell lines in addition to MDA-MB-231 and tested the chemotherapy sensitizer role of LOX inhibition. We demonstrated that inhibiting LOX sensitizes these cells to doxorubicin in collagen I-embedded culture, similar to the results obtained for MDA-MB-231 (new **Supplementary Fig. 7b**). In addition, we tested the sensitizer role of LOX in two ER+ cell lines, MCF-7 and T47D, and observed no increase in doxorubicin sensitivity, suggesting that doxorubicin sensitization conferred by LOX inhibition is potentially specific to TNBC (new **Supplementary Fig. 7c**). This is now included in Page 11, Lines 251-258.

Furthermore, we also extended the chemosensitizer role of LOX inhibition to two other chemotherapy agents than doxorubicin, namely paclitaxel and epirubicin. Paclitaxel is among the most commonly used chemotherapeutics for the treatment of TNBC patients in both neo-adjuvant^{6,7} and adjuvant⁸⁻¹⁰ settings. Epirubicin is a DNA damaging agent belonging to the anthracycline class similar to doxorubicin and is also used for the treatment of TNBCs¹¹. As shown in new **Supplementary Fig. 7d**, LOX inhibition also conferred sensitization to these chemo-agents in collagen I-embedded MDA-MB-231 cells. These results suggest that LOX inhibition could potentially be a general chemotherapy sensitizer in TNBCs. This is now included in Page 11, Lines 258-260.

2. The use of patient-derived xenograft models is lacking and would significantly increase the clinical impact and relevance of the study

Response: We agree with the reviewer that the use of patient-derived xenografts (PDXs) will significantly increase the clinical relevance of our findings. To this end, we first analyzed the gene expression profiling and drug response data of 15 well-characterized TNBC PDX models and observed a significant positive correlation between LOX expression and hypoxia and focal adhesion scores of these PDX tumors (new **Fig. 5a**), suggesting that LOX might be a driver in these processes also in PDXs and hence, increasing the clinical relevance of our findings. Based on these analyses, we selected a TNBC PDX model (TM01278) that is 1.) resistant to doxorubicin (and several other chemotherapy agents), 2.) expresses high levels of LOX, hypoxia and focal adhesion scores and 3.) that is also readily available in Jackson Lab. Then, we generated organoid cultures of this doxorubicin-resistant PDX tumors and demonstrated that our combination treatment significantly decreased organoid size compared to single agent treatments after 6 days of treatment (new **Fig. 5b, c**). More importantly, our *in vivo* experiments showed a significant decrease in tumor growth upon combination therapy in this chemoresistant PDX model expressing LOX (new **Fig. 5d-f**). These results complement our *in vitro* and *in vivo* findings with established cell lines and increase the clinical relevance of LOX inhibition to overcome chemotherapy resistance in TNBC. This is now provided in Pages 14-15, Lines 334-351 in the revised manuscript.

In addition to the PDX organoids, we tested the combination of doxorubicin with BAPN in a treatment-naïve primary TNBC patient organoid, F149T. Combining doxorubicin with BAPN in these patient organoids resulted in a significant reduction in organoid growth (**Fig. 6r, s**), similar to the results obtained with the PDX organoids. This new patient organoid data replaced the previous patient organoid (chemosensitive) data in the original manuscript. These results further support the clinical relevance of LOX inhibition for the treatment of chemoresistant TNBCs. This is now provided in Page 16, Lines 369-372 in the revised manuscript.

3. Fig 1f, i: The effect of ITGA5 and FN1 together should be examine as prognostic factors instead of separately and also confirmed in additional independent human datasets.

Response: As the reviewer suggested, we analyzed the combinatorial effect of ITGA5 and FN1 on patient survival in the KM Plotter database that we used to generate the survival curves for ITGA5 and FN1 separately in the initial version of the manuscript. Combined analysis of ITGA5 and FN1 expressions did not yield a better separation than what we achieved for ITGA5 alone in this dataset, potentially due to an already very significant ($p=0.0008$) effect of ITGA5 alone on patient survival in this dataset. Next, we analyzed an additional independent dataset of the gene expression profiles in 106 chemotherapy-treated TNBC patients as suggested by the reviewer. As a result, we observed that again ITGA5 and FN1 separately yielded a significant difference in overall survival ($p=0.002$, HR=3.53 for ITGA5; and $p=0.004$, HR=3.09 for FN1), supporting our initial analyses results (new **Supplementary Fig. 2b**). Importantly, when the patients are stratified based on ITGA5 and FN1 expressions in combination, as suggested by the reviewer, a stronger separation of overall survival was achieved ($p=0.0008$, HR=5.28 for combination of ITGA5 and FN1, new **Supplementary Fig. 2c**), suggesting that ITGA5/FN1 levels are critical for determining patient outcome in chemotherapy-treated TNBC patients. This is now provided in Page 7, Lines 161-166 in the revised manuscript.

4. LOX expression is missing across the experimental models tested in Figs. 3-5. This precludes subsequent conclusions concerning the causal role of LOX in driving downstream effects.

Response: This is a very well-taken point to demonstrate the causal role of LOX in driving the downstream effects we show. In new **Fig. 3c**, where we compare the activation of downstream signaling

in doxorubicin-treated MDA-MB-231 cells grown in the absence or presence of collagen I, we demonstrated that LOX levels are decreasing upon doxorubicin treatment of cells cultured without collagen I that is in line with the reduction in HIF-1 α protein levels. We further confirmed the critical importance of LOX in mediating doxorubicin resistance by overexpressing LOX open reading frame (ORF) in MDA-MB-231 cells and examining doxorubicin response in the presence of type I collagen. We validated LOX overexpression using both anti-flag and LOX antibodies and showed an increase in ITGA5 levels in these cells (new **Fig. 3e**). Importantly, LOX overexpression conferred resistance to doxorubicin in a dose-dependent manner (new **Fig. 3d, e**). We further showed that inhibiting LOX with two different siRNAs sensitized cells to doxorubicin (new **Fig. 3g**), demonstrating the causal role of LOX in promoting doxorubicin resistance. We also examined the LOX expression in collagen-embedded MDA-MB-231 cells treated with doxorubicin and BAPN combination where we did not see any difference in LOX levels, suggesting that BAPN-induced decrease in LOX activity is not accompanied by a decrease in total protein levels (new **Fig. 3o**). This is now provided in Pages 11, Lines 245-256 in the revised manuscript.

To show the LOX levels/activity in the models used in *in vivo* experiments (new **Figures 4-6**), we performed qRT-PCR, western blot, and IHC staining of LOX as well as LOX activity assay in tumors. We validated the expression of LOX in 4T1 cells as well as tumors and demonstrated that they express LOX at a level similar to the high LOX expressing MDA-MB-231 model (new **Supplementary Fig. 10a, b**). We also demonstrated LOX activity inhibition upon BAPN treatment in MDA-MB-231 xenografts and 4T1 syngeneic tumors (**Fig. 4e, j**). Furthermore, we showed LOX expression and activity in PDX tumors (new **Fig. 5d, g**). Finally, we showed LOX knockdown in 231.shLOX xenografts *via* qRT-PCR, Western blot and IHC (new **Supplementary Fig. 11d, new Fig. 6f-h**) and reduction of LOX activity (new **Supplementary Fig. 11c**). This is now provided in Page 14, Lines 323-325, Page 13, Lines 315-317, Page 14, Lines 326-328, Pages 14-15, Lines 342-346 and Page 15, Lines 359-361 in the revised manuscript.

Overall, thanks to the reviewer, adding these necessary controls in these experiments further supported our findings.

5. Fig 3i: Biochemical data is missing to show FAK and Src inhibitors reduce phospho-levels as was shown in previous panels Fig 3b, e. Clinical implications of the study would be greatly improved if the combinatorial effect of these drugs were validated in animal models.

Response: As the reviewer suggested, we performed Western blot analysis of p-FAK and p-Src in collagen-embedded cells treated with doxorubicin in combination with FAK inhibitor (PF-562271) or Src inhibitor (Saracatinib). We observed that FAK or Src inhibitors effectively reduced the phosphorylation of FAK and Src in combination with doxorubicin (new **Supplementary Fig. 9a**), similar to our initial results with the combination of doxorubicin and BAPN (new **Fig. 3o**). This is now provided in Page 13, Lines 302-304 in the revised manuscript.

In addition, we tested the combination of doxorubicin and FAK/Src inhibitors on tumor growth *in vivo* using 4T1 murine TNBC cells, as suggested by the reviewer. Combining doxorubicin with FAK or Src inhibitors significantly inhibited tumor growth (new **Fig. 4p-s**) and tumor weight (**Supplementary Fig. 10e, f**) as compared to single agent treatments, validating the key roles of FAK and Src in doxorubicin resistance. These data further strengthen the clinical implications of our study. This is now provided Page 14, Lines 330-333 in the revised manuscript.

6. Fig 6c: How was high and low miR-142-3p defined for the KM analysis? The authors should also confirm their findings in additional datasets.

Response: We obtained the median expression of miR-142-3p among all patients in the METABRIC dataset¹² and defined the low miR-142-3p group as patients expressing miR-142-3p at a level lower than the median expression across patients. Conversely, patients were assigned to high miR-142-3p group if they express miR-142-3p at a level higher than the median. Accordingly, we had half of the patients in low miR-142-3p and the other half in high miR-142-3p group. We have included this in the legend of new **Fig. 7c**.

We also analyzed additional patient datasets to further confirm the effect of miR-142-3p on the survival of chemotherapy-treated TNBC patients. These datasets include GSE19536, GSE22220, GSE75685 and GSE40267. However, due to the low number of chemotherapy-treated TNBC patients with miRNA expression profiling (i.e., around 30-40 patients in each), unlike METABRIC dataset from which we were able to stratify 106 chemotherapy-treated TNBC patients with miRNA expressions, we did not get a significant result.

7. The consequences of miR-142-3p on FAK and SRC should be examined *in vitro* and *in vivo*.

Response: As suggested by the reviewer, we performed western blot analysis of p-FAK and p-Src, in miR-negative vs. miR-142-3p transfected MDA-MB-231 cells. miR-142-3p overexpression caused a prominent decrease in the phosphorylation levels of FAK and Src kinases whereas no change was detected in the expression of total FAK and Src proteins (new **Fig. 7i**). Importantly, decreased phosphorylation of FAK and Src as well as downregulation of HIF1A, LOX and ITGA5 protein levels were also validated in MDA-MB-231 xenografts stably expressing miR-142-3p (new **Fig. 7j** and **Supplementary Fig. 12e**). These results suggest that miR-142 overexpression decreases FAK/Src signaling *via* downregulation of LOX, ITGA5 and HIF1A both *in vitro* and *in vivo*. This is now provided in Page 17, Lines 392-395 in the revised manuscript.

8. LOX has been well-reported to have roles in metastasis, the process that drives lethality in breast cancer. The authors should strongly consider experiments/discussion on the impact of targeting LOX and/or downstream components such as FAK/SRC on metastatic biology.

Response: As the reviewer stated, LOX is very well-known for its roles in metastasis. LOX secretion from hypoxic tumors leads to collagen cross-linking at the pre-metastatic site and enhances the accumulation of myeloid-derived suppressor cells, resulting in increased lung metastasis in TNBC models¹³. Moreover, higher LOX expression was observed in metastatic brain tumors of breast cancer patients¹⁴. Furthermore, Baker *et al* demonstrated that LOX increases cell proliferation and invasion in colorectal cancer *via* Src activation¹⁵. The same group has also demonstrated LOX overexpression enhances matrix stiffness and activates FAK, leading to invasion *in vitro* and metastasis *in vivo* in colorectal cancer¹⁶. These findings together with our results suggest that FAK and Src are important downstream effectors of LOX not only in metastasis, but also in drug resistance. This is now included in Page 18, Lines 421-428 in the revised manuscript.

9. It remains unclear the proportion of chemo-resistant patients that may benefit from LOX therapy. This should be included in the discussion.

Response: We thank the reviewer for this insightful comment. It is indeed highly important to determine the proportion of chemoresistant patients that may benefit from LOX inhibition to increase the clinical relevance of our findings. To this end, we first analyzed GSE31519 dataset where we observed that more than half of the patients that exhibit event, described as either relapse or distant metastasis, express higher LOX levels (p-value=0.012) (**Supplementary Fig. 5a**). Accordingly, among 38 chemoresistant TNBC patients, 25 of them has high LOX, suggesting that 66% of chemoresistant TNBC patients are expected to

benefit from LOX inhibition for enhancing chemoresponse. Similarly, in another dataset (GSE25066), we demonstrated that a significantly higher proportion of chemotherapy-treated TNBC patients with residual disease (RD) (56%) express higher levels of LOX (p-value=0.011) (**Supplementary Fig. 5b**). Furthermore, TNBC patients with RD express significantly higher levels of LOX, ITGA5 and FN1 (**Supplementary Fig. 5c-e**). Overall, more than half of the chemoresistant TNBC patients might benefit from LOX inhibition. In addition to these mRNA level analyses, we stained the LOX protein in primary tumor tissues (**Fig. 2p, Supplementary Table 4**) of 77 TNBC patients treated with chemotherapy and showed that higher LOX protein levels were significantly associated with worse DFS (p-value=0.019, **Fig. 2q**), further supporting the translational potential of targeting LOX in clinics. This is now included in the Discussion section of the revised manuscript (Pages 19-20, Lines 461-468).

Reviewer #3 (Remarks to the Author):

In this manuscript the authors report that the ECM remodelling protein LOX is a key player in resistance of triple negative breast cancer (TNBC) to standard chemotherapy through the LOX/ITGA5/FN1/FAK/Src axis. They base the initial hypothesis in a single cell line MDA-MB-231 TNBC cells xenografted in mice, which is poorly representative of TNBC. Among the upregulated pathways in MDA-MB-231 doxorubicin resistant tumors they find integrin signaling leading to increased cell survival. The authors propose that resistant tumors become so by activating hypoxia signaling, which in turn upregulates LOX, ITGA5 and FN1 genes; Saatci et al. They propose that LOX inhibition overcomes resistance both *in vitro* and *in vivo*. They finally propose miR-142-3p as an upstream regulator of the hypoxia-LOX-integrin signaling axis responsible for chemoresistance in TNBC.

The manuscript is not easy to follow and seems a patchwork, where several disconnected results have been pushed together; authors keep jumping from one hypothesis to the next with poor links between them. The MS multiple hypotheses are not fully sustained by the data shown herein and multiple key points would be required in order to functionally demonstrate the involvement of the miR-142-3p-HIF1 α -ITGA5-FN1 in TNBC chemoresistance and to strengthen the links between them.

Response: We thank the reviewer for his/her comments and suggestions he/she made to further strengthen our manuscript. In line with reviewer's comments, we included several other breast cancer cell lines (both TNBC and non-TNBC) and other chemotherapy agents used in clinics to treat TNBCs; new *in vivo* data with well-characterized, chemotherapy resistant patient-derived xenografts (PDXs); new *in vivo* data with FAK and Src inhibitors; new data with 3D PDX-derived and patient-derived organoids; new clinical data; IHC staining of LOX in the tissues of chemotherapy-treated TNBC patients; new biochemical and microscopy data supporting the role of LOX in chemoresistance by quantitatively examining collagen cross-linking, fibronectin assembly and drug penetration in *in vitro* and *in vivo* settings; validation of LOX expression/knockdown/activity and the expression of other LOX family members, e.g. LOXL2 in *in vitro* and *in vivo* experiments; *in vivo* effect of miR-142-3p on HIF1 α /LOX/ITGA5 and downstream FAK/Src signaling; and addressed all the technical concerns of the reviewer. We trust that addition of these necessary experiments and data further supported the role of miR-142-3p/HIF1 α /LOX/ITGA5/FN1 axis in chemoresistance and strengthened the link between each section.

1. It is unclear why the authors compare doxorubicin resistant tumors and "residual disease" to generate the DoxoR-GS. Cells that survive short term doxorubicin treatment are also resistant. How many independent tumors were used for RNAseq. MDA-MB-231 is not representative of clinical TNBC. A

welcome strategy would be to include additional cell lines and Patient derived-xenograft (PDX), which would improve the reliability of the results.

Response: The tumors that we considered “resistant” were treated with doxorubicin for a long time period (i.e., 3 months) which allowed them re-gain proliferative capacity, re-grow and become resistant *via* activation of LOX and integrin signaling. On the other hand, the tumors that we denoted as “sensitive” were shrinking (i.e., they had negative tumor volume fold change) (new **Fig. 1b and c**) at the time that we collected tumors for RNA-Seq analyses. Importantly, we demonstrated the association of our *in vivo*-derived DoxoR-GS with chemo-treated TNBC patient survival in two independent patient datasets (**Fig. 1d** and **Supplementary Fig. 1a**). To further support the clinical relevance of our DoxoR-GS, we analyzed another TNBC patient dataset, GSE31519 in terms of the association of our DoxoR-GS with chemotherapy response. Here, our DoxoR-GS was higher in chemo-treated TNBC patients that exhibit an event, described as either disease relapse or distant metastasis vs. in those with no event (**Supplementary Fig. 1b**). Altogether, we believe that all these results support the comparison we made between resistant and sensitive tumors to generate DoxoR-GS. We sequenced 4 independent tumors from vehicle, sensitive and resistant groups. This is now described in Page 6, Lines 129-131 in the revised manuscript. Furthermore, we uploaded RNA-Seq data to the Sequence Read Archive (SRA) of NCBI with the submission ID: SUB6918779 in accordance with the *Nature Communications* guidelines.

MDA-MB-231 cells are among the most commonly studied TNBC cell line that was first established from a pleural effusion of a 51-year-old Caucasian female with a metastatic mammary adenocarcinoma in the 1970s^{17,18}. However, as the reviewer suggested, we included other TNBC and non-TNBC cell lines in order to further support our results. In this line, we first analyzed LOX expression in a panel of breast cancer cell lines from different molecular subtypes. As shown in new **Supplementary Fig. 7a**, LOX is higher in TNBC cell lines in general compared to HER2-positive and ER+ cell lines. Among different TNBC cell lines, we selected MDA-MB-157 and MDA-MB-436 cell lines in addition to MDA-MB-231 and tested the chemotherapy sensitizer role of LOX inhibition. We demonstrated that inhibiting LOX sensitizes these cells to doxorubicin in collagen-embedded culture, similar to the results obtained for MDA-MB-231 (new **Supplementary Fig. 7b**). In addition, we tested the sensitizer role of LOX in two ER+ cell lines, MCF-7 and T47D, and observed no increase in doxorubicin sensitivity, suggesting that doxorubicin sensitization conferred by LOX inhibition is potentially specific to TNBC (new **Supplementary Fig. 7c**). This is now described in Page 11, Lines 251-258 in the revised manuscript.

In addition, we tested the chemosensitizer role of LOX inhibition in systematically selected TNBC PDXs and in PDX- as well as TNBC patient-derived organoids. To this end, we first analyzed the gene expression profiling and drug response data of 15 well-characterized TNBC PDX models and observed a significant positive correlation between LOX expression and hypoxia and focal adhesion scores of these PDX tumors (new **Fig. 5a**), suggesting that LOX might be a driver in these processes also in PDXs and hence, increasing the clinical relevance of our findings. Based on these analyses, we selected a TNBC PDX model (TM01278) that is 1.) resistant to doxorubicin (and several other chemotherapy agents), 2.) expresses high levels of LOX, hypoxia and focal adhesion scores and 3.) that is also readily available in Jackson Lab. Then, we generated organoid cultures of this doxorubicin resistant PDX tumors and demonstrated that LOX inhibition in combination with doxorubicin significantly decreased organoid size compared to single agent treatments after 6 days of treatment (new **Fig. 5b, c**). More importantly, our *in vivo* experiments showed a significant decrease in tumor growth upon combining LOX inhibitor with doxorubicin in this chemoresistant PDX model expressing LOX (new **Fig. 5d-f**). These results complement our *in vitro* and *in vivo* findings with established cell lines and increase the clinical relevance

of LOX inhibition to overcome chemotherapy resistance in TNBC. This is now provided in Pages 14-15, Lines 334-351 in the revised manuscript.

In addition to the PDX organoids, we also tested the combination of doxorubicin with BAPN in a treatment-naïve primary TNBC patient organoid, F149T. Combining doxorubicin with BAPN in these patient organoids resulted in a significant reduction in organoid growth similar to results achieved with the PDX organoids (**Fig. 6r, s**). This new patient organoid data replaced the previous patient organoid (chemosensitive) data in the original manuscript. These results further support the clinical relevance of LOX inhibition for the treatment of chemoresistant TNBCs. This is now provided in Page 16, Lines 369-372 in the revised manuscript.

2. In Fig 1j authors claim a significant association between high FN1 and poorer RFS in chemotherapy-treated basal patients, but $p = 0.05$, indicating no significance. Moreover, Are the chemotherapy-treated TNBC patient datasets in which the DoxoR-GS score was analyzed from patients with analogous clinical management? This might be relevant since the tumors in mice were homogenously generated using doxorubicin and not with other chemotherapies such as platinum, taxanes, or olaparib. It would be relevant to know whether the proposed LOX-mediated mechanism is involved in chemoresistance beyond doxorubicin.

Response: We apologize for the mistake we made on giving the exact p-value. The correct p-value should be $p=0.049$ as it is provided by the KM plotter website (kmplot.com/analysis). In order to further verify the role of FN1 expression on predicting survival in chemotherapy-treated TNBC patients, we analyzed an additional independent dataset of the gene expression profiles in 106 chemotherapy-treated TNBC patients. As a result, we observed that again ITGA5 and FN1 separately yielded a significant difference in overall survival ($p=0.002$, HR=3.53 for ITGA5; and $p=0.004$, HR=3.09 for FN1), supporting our initial analyses results (new **Supplementary Fig. 2b**). Importantly, when the patients are stratified based on ITGA5 and FN1 expressions in combination, a stronger separation of overall survival was achieved ($p=0.0008$, HR=5.28 for combination of ITGA5 and FN1, new **Supplementary Fig. 2c**), suggesting that ITGA5/FN1 levels are critical for determining patient outcome in chemotherapy-treated TNBC patients. This is now described in Page 7, Lines 15-167 in the revised manuscript.

In the dataset (GSE58812) that we used to generate the DoxoR-GS, treatment regimen information has unfortunately not been provided. Despite the lack of detailed treatment information, it is known that TNBC patients are mostly treated with a cocktail of different chemotherapy agents¹⁹. Moreover, the KM Plotter database that we used for survival analyses of chemotherapy-treated TNBC patients (**Fig. 1h, k, and 2o**) includes patients treated with several different chemotherapy agents. Furthermore, we stained the LOX protein in primary tumor tissues (**Fig. 2p, Supplementary Table 4**) of 77 TNBC patients treated with chemotherapy and showed that higher LOX protein levels (median separated) were significantly associated with worse disease-free survival (DFS) (p -value=0.019, **Fig. 2q**). There, 35% of the patients received anthracycline-based therapy while 43% received anthracyclines in combination with taxanes, and 22% received other chemotherapy agents. Therefore, this suggests ~~LOX~~ LOX expression is associated with response to different chemotherapy agents, and its targeting may sensitize cells to chemotherapy. As an experimental proof of LOX inhibition-mediated sensitization to different chemotherapy agents, we tested the combination of BAPN with two different chemotherapy agents other than doxorubicin: paclitaxel and epirubicin. Paclitaxel is among the most commonly used chemotherapeutics for the treatment of TNBC patients in both neo-adjuvant^{6,7} and adjuvant⁸⁻¹⁰ settings. Epirubicin is a DNA damaging agent belonging to the anthracycline class similar to doxorubicin and is also used for the treatment of TNBCs¹¹. As shown in new **Supplementary Fig. 7d**, LOX inhibition also conferred sensitization to these chemo-agents in collagen I-embedded MDA-MB-231 cells. These results

suggest that LOX inhibition could potentially be a general chemotherapy sensitizer in TNBCs. This is now provided in Page 11, Lines 258-260.

3. Have the authors checked the levels of LOX, ITGA5 and FN1 in the vehicle tumors generated in mice from 231 cells, to demonstrate that these genes are only overexpressed in chemoresistant tumors vs sensitive ones?

Response: We compared the expression levels of LOX, ITGA5 and FN1 in vehicle, doxorubicin-sensitive and doxorubicin-resistant tumors (n=4 for each group) in our RNA-seq data as suggested by the reviewer. As shown in new **Supplementary Fig. 3a**, LOX, ITGA5 and FN1 mRNAs are significantly upregulated only in resistant tumors compared to sensitive ones. This is now provided in Page 8, Lines 191-194.

4. In Fig. 2g the authors should check FAK and Src total levels in order to conclude that their phosphorylation levels are upregulated. The same is true for the blots in Fig. 2j, 3b and 3e. Is this LOX upregulation also seen in 231 cells upon doxorubicin treatment? Have the authors attempted to overexpress LOX in 231 cells and check for their sensitivity towards doxorubicin? That would help sustain LOX claimed role.

Response: We added blots for total protein levels of FAK and Src to new **Fig. 2j** (hypoxia) and **2m** (siLOX) and **Fig. 3c** (no collagen vs. collagen-embedded culture) and **3o** (combination of doxorubicin with BAPN in collagen-embedded culture) in the revised version of the manuscript. The total protein levels of FAK and Src did not change in new **Fig. 2j, m** and **Fig. 3c, o**, confirming our initial claims.

We checked LOX expression in uncoated vs. collagen-embedded cells with or without doxorubicin treatment. We observed that LOX levels are decreasing upon doxorubicin treatment of cells cultured without collagen that is in line with the reduction in HIF1A protein levels (new **Fig. 3c**). It has previously been shown that doxorubicin reduces HIF1A transcriptional activity that further reduces LOX levels and subsequent cell migration²⁰. Here, when we embedded cells in type I collagen matrix, HIF-1 α is stabilized which was accompanied by increased LOX expression and subsequent activation of downstream signaling, leading to chemoresistance. This is now described in Page 11, Lines 248-252.

We further confirmed the critical importance of LOX in mediating doxorubicin resistance by overexpressing LOX open reading frame (ORF) in MDA-MB-231 cells and examining doxorubicin response in the presence of type I collagen. We validated LOX overexpression using both anti-flag and LOX antibodies as well as showed the increase in ITGA5 levels in these cells (new **Fig. 3e**). Importantly, LOX overexpression conferred resistance to doxorubicin in a dose-dependent manner (new **Fig. 3d**). We further showed that inhibiting LOX with two different siRNAs sensitized cells to doxorubicin (new **Fig. 3g**). Overall, these data demonstrate the causal role of LOX in promoting doxorubicin resistance. This is now described in Pages 11, Lines 248-256.

5. In figure 2j phosphorylation of FAK and Src is downregulated when LOX is inhibited, however, this downregulation is not observed in the presence of the LOX inhibitor BAPN in Figure 3e. Authors should explain these discrepancies. Moreover, LOX silencing as shown in Fig. 2j (32kDa) is too weak. A stable or stronger silencing is required to conclude it has a downstream impact.

Response: We thank the reviewer for raising this important point that we did not adequately explain in the initial version of the manuscript. In new **Fig. 3o** where we do not see inhibition of downstream FAK/Src signaling upon LOX inhibition alone, we cultured cells in the presence of collagen I, the major substrate of LOX, unlike cells in new **Fig. 2m**, where we do not use exogenous LOX substrate (cells

were in 2D culture). When we inhibited LOX in cells cultured in the absence of collagen, we observed that ITGA5 and FN1 are downregulated, leading to reduced FAK/Src phosphorylation (new **Fig. 2m**). Furthermore, we showed that doxorubicin treatment decreases FAK and Src phosphorylation in cells grown in the absence of collagen (new **Fig. 3c**). On the other hand, phosphorylation levels are preserved in collagen-embedded cells, suggesting that LOX activity counteracts with doxorubicin-mediated decrease in FAK/Src signaling. When LOX is inhibited with BAPN in combination with doxorubicin, collagen cross-linking and fibronectin assembly are reduced, leading to increased drug penetration into cancer cells (new **Fig. 3h-n**). This further enhances the effect of doxorubicin on FAK/Src signaling and promotes drug-induced cell death (new **Fig. 3o, p**). We further validated these results in different tumor models *in vivo*, where LOX inhibition with BAPN decreased LOX activity and collagen cross-linking, and enhanced doxorubicin penetration into tumors, leading reduced downstream FAK/Src signaling specifically in combination-treated tumors (new **Fig. 4, 5**). This is now discussed in Pages 18-19, Lines 439-445 in the revised manuscript.

In order to improve LOX knockdown, we used two different inducible shRNAs targeting LOX and observed strong downregulation of LOX as well as its downstream ITGA5 and FN1 and FAK/Src signaling (**Fig. 2n**), supporting the proposed role of LOX. This is now provided in Pages 9, Lines 215-217.

6. LOX activity is not easy to measure, the data shown in the graph 2h lacks negative controls (such as adding BAPN inhibitor). How is LOX activity in resistant tumors? That would be more interesting than analyzing it in cell culture.

Response: In the revised version of the manuscript, we measured the LOX activity in hypoxic cells, with the addition of BAPN-treated cells as a negative control. As shown in new **Fig. 2k** of the revised manuscript, hypoxia doubles the LOX activity whereas BAPN treatment reduces it to half. We also measured LOX activity in doxorubicin-resistant tumors from **Fig. 4a** of the revised manuscript, and further demonstrated that addition of BAPN to doxorubicin causes a significant decrease in LOX activity as expected (new **Fig. 4e**). In addition, we measured the LOX activity in BAPN-treated PDX tumors (new **Fig. 5g**). Finally, we analyzed LOX activity in our 231.shLOX tumors and demonstrated that LOX activity was decreased upon LOX knockdown (**Supplementary Fig. 11c**). This is now provided in Page 9, Lines 208-210, Page 13, Lines 315-317, Page 15, Lines 346-347 and Page 15, Lines 359-361.

7. BAPN is not a selective inhibitor of LOX since it also has activity against other LOX family members. The authors have not checked the levels of those proteins, particularly LOXL2, in 231 cells, where it has been shown to be expressed and have a prominent role (Moreno-Bueno et al., 2011). LOX levels should also be shown.

Response: We agree with the reviewer that BAPN is not a selective inhibitor of LOX, which we had mentioned in the discussion of the initial manuscript. We also agree that we need to show the expression of our target in MDA-MB-231 cells. To this end, we first examined the expression of LOX in these cells and observed that LOX is expressed at high levels in this cell line among different breast cancer cell lines (new **Supplementary Fig. 7a**). We also examined LOX levels in MD-MB-231 xenografts *via* qRT-PCR, Western blotting and IHC (new **Supplementary Fig. 11d** and **Fig. 6f-h**). Furthermore, we had shown in the initial manuscript that whereas LOX is significantly upregulated in our doxorubicin-resistant xenografts, LOXL2 or other family members do not demonstrate a differential expression (**Supplementary Fig. 3b**). Then, we analyzed the expression of LOXL2 and other LOX family members in our 231.shLOX tumors to further verify the LOX-specific effects on doxorubicin response. We observed a significant decrease in LOX expression/activity in the shLOX tumors (new **Fig. 6f-h and**

Supplementary Fig. 11c, d) whereas no decrease in the expression of LOXL2, or other LOX family members were detected (new **Supplementary Fig. 11e, f**). Importantly, we observed that LOX overexpression confers resistance to doxorubicin (new **Fig. 3d, e**) while its knockdown with two different siRNAs resulted in doxorubicin sensitization (new **Fig. 3g**). All these data suggest that doxorubicin sensitization is majorly driven by inhibition of LOX and is independent of other LOX family members, at least in our TNBC models.

8. What is the effect on LOX and HIF1alpha expression in cells cultured with doxorubicin in the presence of the absence of collagen? (Figure 3b).

Response: As mentioned above, we checked LOX expression in uncoated vs. collagen-embedded cells with or without doxorubicin treatment. We observed that LOX levels are decreasing upon doxorubicin treatment of cells cultured without type I collagen that is in line with the reduction in HIF1A protein levels (new **Fig. 3c**). It has previously been shown that doxorubicin reduces HIF1A transcriptional activity that further reduces LOX levels and subsequent cell migration²⁰. Here, when we embedded cells in collagen I matrix, HIF-1 α is stabilized that was accompanied by increased LOX expression and subsequent activation of downstream signaling, leading to chemoresistance. This is now provided in Pages 11, Lines 247-248.

9. The graph on the right of images in Fig. 3g is not properly labelled and does not seem to account for the differences displayed by the images. Besides, these immunofluorescence images cannot account as a formal method to analyse collagen and FN deposition. The same is true for the experiments shown in Fig. 3h. The results displayed in Fig. 3g and 3h are not adequate to back up the authors' conclusion regarding "a prominent decrease in collagen cross-linking" or "significant reduction in fibronectin deposition" (page 15). These are few images that might not be representative. Additional images should be acquired per experiment with replicates, and intensity quantified using adequate software (i.e. as performed in Qureshi et al, Biol Open 2017) and an alternative method is required. For Figure 3h, statistical analysis is required to state that there is a strong co-localization of type I collagen and fibronectin fibrils?

Response: The quantification that we provided for the decrease in collagen cross-linking and fibronectin assembly upon LOX inhibition under collagen-embedding (new **Fig. 3h**) was done based on the results of three independent experiments. In the revised version of the manuscript, we corrected the labeling and replaced the representative images with new ones that better reflect the quantitative data (new **Fig. 3i**). In addition, we renewed the images from the HFF-derived ECM experiment that now better shows the positioning of cancer cells within the ECM and also added the scale bars (50 μ m). We also quantified collagen and fibronectin intensities in Fig. 3j (ECM produced by HFF cells) based on the results of three independent experiments to justify the reduction in cross-linked collagen (represented as thick collagen fibers), as well as the extracellular deposition and assembly of fibronectin upon LOX inhibition (new **Fig. 3k**). The quantification was done using the ImageJ software that is commonly used to analyze imaging data⁴. This is now provided in Page 12, Lines 269-281.

In addition to re-analyzing the images and providing quantifications, we also performed additional biochemical assays to demonstrate the decrease in collagen cross-linking and fibronectin deposition and assembly as suggested by the reviewer. For collagen cross-linking, we utilized the collagen assay from Abcam (ab222942) that quantifies the amount of hydroxyproline residues present in cross-linked collagen fibers. We observed that the amount of cross-linked collagen in HFF-derived ECM that was incubated with BAPN-treated MDA-MB-231 cells was significantly less compared to the ECM incubated with vehicle-treated cells (new **Fig. 3l**). For fibronectin fibril assembly and deposition, we performed deoxycholate (DOC) lysis, which is a standard biochemical method to separate the secreted

form of fibronectin from the assembled form within the ECM²¹⁻²⁴, using both HFF- and NIH-3T3-cell derived ECM. Western blot analysis of fibronectin in soluble and non-soluble fractions demonstrated that BAPN treatment causes a decrease in both the deposited soluble fibronectin as well as the assembled fibronectin (new **Fig. 3m**). These results further confirmed the immunofluorescence staining results, showing a reduction in fibronectin deposition and assembly upon LOX inhibition. This is now provided in Page 12, Lines 286-290.

To quantify the co-localization between collagen and fibronectin fibers in “only ECM”, “vehicle” and “BAPN” groups in new **Fig. 3j**, we measured the spatial overlap between the labels from different color channels in identical pixel positions in each image using a macro in the ImageJ software⁴. As a result, we detected a significant decrease in the percentage of collagen fibers co-localized with fibronectin upon LOX inhibition (new **Supplementary Fig. 8a**). This is now provided in Page 12, Lines 281-283.

Overall, these results support our conclusion with respect to LOX inhibition-mediated decrease in collagen cross-linking or fibronectin deposition.

10. Moreover, in Fig. 3h there are no clear differences in vehicle vs BAPN in the images displayed. In this line in page 10, second paragraph: with the reported results, the authors cannot state “...demonstrated the reversal of collagen cross-linking upon LOX inhibition...”. For that, further biochemical experiments are required.

Response: As stated above, we provided quantifications for collagen and fibronectin intensities and replaced the images with new ones that better represent the quantifications in new **Fig. 3h** and **3j**. In addition, we performed the biochemical collagen assay that is based on hydrolysis of cross-linked collagen peptides to generate hydroxyproline which is then converted into a brightly colored chromophore that can be detected at OD 560 nm. As a result, we detected a significantly lower amount of cross-linked collagen in HFF-derived ECM that was incubated with BAPN-treated MDA-MB-231 cells (new **Fig. 3l**). This is now provided in Page 12, Lines 269-278, 286-290.

11. In Figure 3i authors show the percentage of growth inhibition of doxorubicin in the presence of FAK and Src inhibitors, but what is the percentage of apoptotic cells? This needs to be clarified since authors conclude: “LOX-mediated collagen cross-linking and the subsequent increase in fibronectin fibril assembly contribute to chemotherapy resistance via activation of FAK/Src signaling and inhibition of apoptosis in chemotherapy-treated TNBCs”.

Response: As suggested by the reviewer, we performed Cleaved Caspase-3/phalloidin staining to demonstrate the percentage of apoptotic cells embedded in collagen and treated with doxorubicin in combination with FAK or Src inhibitors. As a result, we observed a significant increase in the percentage of apoptotic cells upon combination of doxorubicin with FAK and Src inhibitors (from 10% in Doxo group to 30% in Doxo+PF-562271 and to 40% in Doxo+Saracatinib groups, new **Supplementary Fig. 9b, c**). This is now provided in Page 13, Lines 302-304.

12. In 4T1 cells the authors should check the levels of LOX and LOX family members, at least LOXL2, since the effects observed upon BAPN treatment could be due to LOXL2 inhibition. It might be possible that LOX is not expressed at all in these tumors and the LOX activity shown in Fig. 4h is not due to LOX itself.

Response: To demonstrate that our target, LOX is expressed in 4T1 tumors, we examined its levels in 4T1 cells and tumors by qRT-PCR and western blot analysis in comparison with MDA-MB-231, which we demonstrated to express high levels of LOX (please see our answer to the comment #7). We observed

that 4T1 cells express similar levels of LOX protein with MDA-MB-231 cells (new **Supplementary Fig. 10a**). 4T1 syngeneic tumors also express similar levels of LOX mRNA and protein levels with the MDA-MB-231 tumors (new **Supplementary Fig. 10b**). This is now provided in Page 14, Lines 323-325. We also checked the expression of LOXL2 and observed a similar expression level as LOX in 4T1 tumors as shown above (**Figure for Reviewer**).

Figure for Reviewer. qRT-PCR analysis of LOX and LOXL2 in 4T1

13. Figure 4. How the authors explain the combinatory effect of doxorubicin and BAPN in the chemoresistant TNBC model when the inhibition of LOX activity is similar with BAPN and doxorubicin plus BAPN? Another alternative pathway has to be activated in order to explain the combinatory mechanism.

Response: This is a very well-taken comment which helped us a lot to improve our manuscript (this is also related to reviewer's comment #5). As the reviewer has pointed out, in the chemoresistant 4T1 model, we demonstrated that although LOX is inhibited in all BAPN-treated tumors (new **Fig. 4j**), including the BAPN alone and BAPN+Doxo groups, the decrease in tumor growth (new **Fig. 4i**) and effective blockade of downstream FAK/Src signaling (new **Fig. 4o**) was only observed in Doxo+BAPN-treated tumors. It has been proposed that LOX may reduce drug penetration under hypoxic conditions and lessen the cytotoxicity of chemo-agents in 3D collagen cultured cells and tumor models²⁵. We showed that doxorubicin decreased the activation of FAK/Src signaling in 2D cultured cells, but not in 3D collagen culture (new **Fig. 3c**). Therefore, we tested if targeting LOX with BAPN increases doxorubicin penetration and decreases FAK/Src signaling, leading to chemosensitization both *in vitro* and *in vivo* as a mechanism explaining the combinatorial effect of doxorubicin and BAPN.

First, doxorubicin autofluorescence was quantified at an excitation-emission wavelength of 495 nm-595 nm as reported before³ in vehicle, BAPN, doxorubicin and doxorubicin+BAPN treated-cells embedded in type I collagen. We observed significant increase in intracellular doxorubicin in combination-treated cells (**Fig. 3n**). This was accompanied by a decrease in FAK and Src phosphorylation (**Fig. 3o**) and apoptosis (new **Fig. 3p** and **Supplementary Fig. 6d-g**). Then, we moved on to the *in vivo* experiments. Picosirius red staining confirmed the suppression of collagen cross-linking in all BAPN-treated tumors (new **Fig. 4k, l**), similar to the LOX activity (**Fig. 4j**) as it was pointed out by the reviewer. Importantly, effective inhibition of FAK/Src signaling upon combination therapy was confirmed with Western blotting (new **Fig. 4o**), which was accompanied by increased penetration of doxorubicin in the combination-treated tumors (new **Fig. 4m, n**). We then tested this hypothesis also in our PDX model. There, we demonstrated reduced LOX activity (new **Fig. 5g**) and fibrillar collagen content upon LOX inhibition with BAPN in all BAPN-treated groups (new **Fig. 5h, i**), similar to the 4T1 model. Importantly, this was accompanied by enhanced drug penetration into tumors (new **Fig. 5j, k**) and better inhibition of downstream FAK/Src signaling in combination-treated tumors (new **Fig. 5l**). Therefore, we propose that

hypoxia-driven LOX expression, on one hand, leads to increased collagen cross-linking, fibronectin assembly and decreased drug penetration, and on the other hand, increases ITGA5 and FN1 expression, collectively leading to increased FAK/Src signaling, decreased apoptosis and chemoresistance (new **Fig. 8**). This is now explained in detail in Page 19, Lines 441-449 and Page 21, Lines 491-496.

14. The results shown in Fig. 5 are not sound unless LOX protein levels are shown upon shRNA interference. What is the effect of combinatory treatment on pFAK? And the percentage of apoptotic cells?

Response: We performed qRT-PCR (new **Fig. 6f**) and Western blot analysis (new **Supplementary Fig. 11d**) and IHC staining (new **Fig. 6g, h**) of LOX in shCtrl and shLOX tumors and validated LOX knockdown. Furthermore, we measured LOX activity and demonstrated a significant reduction upon LOX knockdown with shRNA (**Supplementary Fig. 11c**). This is now provided in Page 15, Lines 359-361.

Furthermore, we examined p-FAK levels in these MDA-MB-231 xenografts as requested by the reviewer. Since our p-FAK antibody was not suitable for IHC, we performed western blot analysis and demonstrated the decrease in FAK phosphorylation in combination-treated tumors (new **Fig. 6q**). This is now provided in Page 16, Lines 367-368.

We also quantified the percentage of Cleaved Caspase-3 positive cells in all 4 groups and observed the highest percentage in combination-treated tumors (i.e., doxorubicin: 19.5+/- 1.8%, shLOX: 15.4+/- 9.0% and Doxorubicin+shLOX: 39.6+/- 0.6%).

15. Figure 6b. Is there a p-value for the strong inverse correlation between the miRNAs miR-142-3p and miR-128-3p and HIF1alpha, LOX and ITGA5 signatures? Figure 6i. What is the effect of miR-142-3p on pFAK and pSrc?

Response: We provided a table including the p-values for the Pearson correlation analysis between miR-142-3p and HIF1A gene signature score, LOX and ITGA5 expressions (new **Supplementary Fig. 12b**) and between miR-128-3p and HIF1A gene signature score, LOX and ITGA5 expressions (new **Supplementary Fig. 12c**). This is now described in Pages 16, Lines 381-383.

We also examined the levels of p-FAK and p-Src in miR-142-3p-overexpressing cells as shown in new **Fig. 7i**. Accordingly, miR-142-3p overexpression reduces the activation of FAK/Src signaling in parallel with downregulation of LOX, ITGA5 and HIF1A. Importantly, inhibition of phosphorylation of FAK and Src as well as downregulation of HIF1A, LOX and ITGA5 protein levels were also validated in MDA-MB-231 xenografts stably expressing miR-142-3p (new **Fig. 7j** and **Supplementary Fig. 12e**). This is now provided in Page 17, Lines 390-393.

16. The authors claim that LOX might be triggering ITG5A and FN1 expression without formally proving it and at the same time they propose that is miR-142-3p the one that regulates their expression. There are not enough data in the manuscript to sustain these links.

Response: Based on reviewer's comment, we now provided additional data supporting the proposed links between LOX and FAK/Src signaling that involves regulation of ITGA5/FN1 expression, ECM remodeling and changes in drug penetration. The regulation of ITGA5/FN1 by LOX was supported by extending the loss-of-function experiment that was performed with siRNAs to shRNA-mediated knockdowns. Silencing LOX with two different shRNAs resulted in a prominent decrease in both ITGA5 and FN1 expressions in line with the siRNA-mediated knockdown (new **Fig. 2n**). This was further validated *in vivo* where we showed decreased expression of ITGA5 in tumors with LOX knockdown (new **Supplementary Fig. 11d**). Importantly, the mRNA levels of ITGA5/FN1 were strongly positively correlating with LOX expression in TNBC patients (**Fig. 2h**). Conversely, LOX overexpression in MDA-MB-231 cells increased the levels of ITGA5 (new **Fig. 3e**). As mentioned in the text (Page 5, Lines 108-110), LOX has recently been proposed to regulate gene transcription. Boufraquech *et al* demonstrated that LOX knockdown reduces SNAI2 expression and that LOX and SNAI2 expressions correlate in breast cancer tumors. Experimentally, they showed binding of LOX to SNAI2 promoter, activating its transcription²⁶. Similarly, we have observed downregulation of ITGA5/FN1 expression upon LOX knockdown with two different siRNAs and shRNAs, and a very strong positive correlation between LOX and ITGA5 or FN1 mRNA levels in 9 different patient datasets. We also performed ITGA5 promoter assay in MDA-MB-231 and HEK-293 cells overexpressing human and mouse ORF and observed a significant increase in ITGA5 promoter activity upon LOX overexpression (**Figure for reviewer** below, suggesting that LOX is directly or indirectly involved in transcription of ITGA5. We are currently investigating how LOX regulates gene expression, including that of ITGA5 and FN1, using several high-throughput approaches that we hope to publish in future.

Figure for Reviewer. ITGA5 promoter activity upon overexpression of human or mouse LOX ORF in MDA-MB-231 (a) and HEK-293 (b) cell lines.

With respect to the role of miR-142-3p in regulating the expressions of LOX and ITGA5, we demonstrated direct binding of miR-142-3p to the 3'-UTR regions of *HIF1A*, *LOX* and *ITGA5* by luciferase assays (new **Fig. 7l**). We also showed the downregulation of the targets at both mRNA and protein levels upon miR-142-3p overexpression (new **Fig. 7h** and **7i**). For the revised version, we added pFAK and p-Src blots to miR-142 overexpression western blot which further supports the involvement of miR-142-3p in regulating FAK/Src signaling by targeting HIF1A/LOX/ITGA5 axis (new **Fig. 7i**). Furthermore, we demonstrated the downregulation of HIF1A, LOX and ITGA5 protein levels as well as phosphorylation of FAK and Src in MDA-MB-231 xenografts stably expressing miR-142-3p (new **Fig. 7j** and **Supplementary Fig. 12e**), showing that HIF1A, LOX and ITGA5 are regulated by miR-142-3p also *in vivo*. We trust that addition of several new data during the revision in line with the reviewer's comments strengthened these links.

17. In the Discussion, the authors claim that their results “underpin the intense communication between resistant tumor cells and surrounding ECM” (page 15) which is an overstatement given the fact that their results derived mostly from cells *in vitro* cultured and they do not show any ECM modification within the tumors.

Response: We thank the reviewer for suggesting measuring ECM modifications within the tumors. To corroborate our *in vitro* findings on the regulation of collagen cross-linking by LOX in chemoresistance *in vivo*, we performed picrosirius red staining of 1.) our doxorubicin-sensitive and -resistant tumors, 2.) doxorubicin-resistant MDA-MB-231 tumors treated with Doxo+BAPN combination, 3.) combination-treated 4T1 syngeneic tumors, and 4.) chemoresistant PDX tumors treated with Doxo+BAPN. As a result, we detected a significantly higher levels of fibrillar collagen in our doxorubicin-resistant tumors (new **Fig. 2f**) that was downregulated upon LOX inhibition with BAPN (new **Fig. 4f**). The reduction in cross-linked collagen upon LOX inhibition was also observed in the syngeneic 4T1 model (new **Fig. 4k**) as well as in chemoresistant PDXs (new **Fig. 5h**). In addition, we also performed the biochemical collagen assay in these tumors and demonstrated that LOX inhibition in doxorubicin resistant MDA-MB-231 tumors reduced the amount of cross-linked collagen (new **Fig. 4h**). Overall, these new results further validate our findings in *in vivo* settings on the role of LOX in ECM modulation in chemoresistant TNBC tumors.

In summary, although Saatci and co-workers propose an interesting mechanism, they do not demonstrate that LOX is indeed responsible for the TNBC chemoresistance, and the same occurs with most claims. Furthermore, their conclusions are not clear since they mention several mechanisms such as hypoxia, enhanced ECM stiffness or LOX upregulation and subsequent downstream signaling. None of this is formally proven within the tumors.

Response: We believe that we addressed all the concerns/suggestions of the reviewer and justified our proposed mechanism of chemoresistance in TNBCs that involve *hypoxic upregulation of LOX leading to both enhanced collagen cross-linking/reduced drug penetration and ITGA5/FN1 expression, culminating in activation of downstream FAK/Src signaling*. In this line, we showed the expression of LOX and other LOX family members (e.g. LOXL2) in cells and tumors and validated the chemosensitizer role of LOX inhibition in several different TNBC cell line models. We showed that overexpression of LOX confers doxorubicin resistance while its knockdown sensitizes cells to chemotherapy. We validated the sensitizer role of LOX inhibition in well-characterized chemoresistant PDX- and patient-derived organoids, and in PDX models *in vivo*. We validated the effect of LOX on ITGA5 expression and the reduction of downstream FAK/Src signaling upon LOX inhibition in combination with chemotherapy *in vivo*. We further showed that in addition to decreasing integrin signaling *via* downregulation of ITGA5/FN1, targeting LOX leads to reduced collagen cross-linking and enhanced drug penetration and chemosensitization. Importantly, all these hypotheses were tested using different *in vivo* models by performing both staining and biochemical assays. We performed *in vivo* testing of doxorubicin in combination with FAK and Src inhibitors and achieved a similar degree of sensitization as the combination of LOX inhibitor with doxorubicin, further supporting the involvement of these downstream kinases in LOX-mediated chemoresistance. Finally, we added more clinical data (response and prognosis) and performed LOX protein staining in chemotherapy-treated TNBC patient samples and validated the key role of LOX in driving chemoresistance in TNBC.

Overall, thanks to the reviewer, we trust that adding the necessary controls in our experiments to show specific chemoresistance driver role of LOX in TNBCs and validating the proposed axis with new *in vitro*, *in vivo* and patient data further supported our findings.

MINOR POINTS

Revise Figure legends (ie. In Fig. 3g the graph on the right is not mentioned within the text nor the axes are not properly labelled)

Response: In the revised version of the manuscript, we properly labelled the graph in new **Fig. 3i**. We also thoroughly revised all the figure legends.

In Fig. 4f the images displayed suggest an increase in size for Doxo treated tumors vs Ctrl.

Response: In the revised version of the manuscript, we repeated this *in vivo* experiment (vehicle, doxorubicin, BAPN and doxorubicin+BAPN) with a new batch of 4T1 cells (*with higher proliferation capacity*) while we tested the effect of FAK or Src inhibitor on doxorubicin response. As shown in the figure above, we again did not observe any increase in tumor volume or tumor weight in doxorubicin-treated 4T1 syngeneic tumors (**Figure for Reviewer** below), similar to our results in the initial version of the manuscript (now **Fig. 4i** and **Supplementary Fig. 10c, d**).

Figure for Reviewer. Tumor volume (a), tumor weight and tumor pictures (c) of 4T1 syngeneic model upon treatment with doxorubicin and BAPN, alone or in combination.

The journal titles in the reference should be corrected.

Response: We carefully updated the journal titles in the references in the revised manuscript.

Doxorubicin resistant xenografts” should be “doxorubicin-resistant xenografts”.

Response: In the revised version of the manuscript, we have changed all “doxorubicin resistant” to “doxorubicin-resistant”.

References:

1. Azimi, I., Petersen, R.M., Thompson, E.W., Roberts-Thomson, S.J. & Monteith, G.R. Hypoxia-induced reactive oxygen species mediate N-cadherin and SERPINE1 expression, EGFR signalling and motility in MDA-MB-468 breast cancer cells. *Sci Rep* **7**, 15140 (2017).
2. Raj, Y., *et al.* Evaluation of the Nature of Collagen Fibers in KCOT, Dentigerous Cyst and Ameloblastoma using Picrosirius Red Stain - A Comparative Study. *J Clin Diagn Res* **9**, ZC01-04 (2015).

3. Liang, P.C., *et al.* Doxorubicin-modified magnetic nanoparticles as a drug delivery system for magnetic resonance imaging-monitoring magnet-enhancing tumor chemotherapy. *Int J Nanomedicine* **11**, 2021-2037 (2016).
4. Schindelin, J., *et al.* Fiji: an open-source platform for biological-image analysis. *Nat Methods* **9**, 676-682 (2012).
5. Riffo-Campos, A.L., Riquelme, I. & Brebi-Mieville, P. Tools for Sequence-Based miRNA Target Prediction: What to Choose? *International journal of molecular sciences* **17**(2016).
6. Omarini, C., *et al.* Neoadjuvant treatments in triple-negative breast cancer patients: where we are now and where we are going. *Cancer Manag Res* **10**, 91-103 (2018).
7. Senkus, E., *et al.* Primary breast cancer: ESMO Clinical Practice Guidelines for diagnosis, treatment and follow-up. *Ann Oncol* **26 Suppl 5**, v8-30 (2015).
8. Harper, N.W., *et al.* Adjuvant Treatment of Triple-Negative Metaplastic Breast Cancer With Weekly Paclitaxel and Platinum Chemotherapy: Retrospective Case Review From a Single Institution. *Clin Breast Cancer* **19**, e495-e500 (2019).
9. Muss, H.B., *et al.* Adjuvant chemotherapy in older women with early-stage breast cancer. *The New England journal of medicine* **360**, 2055-2065 (2009).
10. Sparano, J.A., *et al.* Weekly paclitaxel in the adjuvant treatment of breast cancer. *The New England journal of medicine* **358**, 1663-1671 (2008).
11. Sun, W., *et al.* Prognostic analysis of triple-negative breast cancer patients treated with adjuvant chemotherapy of fluorouracil, epirubicin and cyclophosphamide. *Oncol Lett* **11**, 2320-2326 (2016).
12. Curtis, C., *et al.* The genomic and transcriptomic architecture of 2,000 breast tumours reveals novel subgroups. *Nature* **486**, 346-352 (2012).
13. Erler, J.T., *et al.* Hypoxia-induced lysyl oxidase is a critical mediator of bone marrow cell recruitment to form the premetastatic niche. *Cancer cell* **15**, 35-44 (2009).
14. Cha, Y.J., Jung, W.H. & Koo, J.S. Site-specific expression of amine oxidases in breast cancer metastases. *Tumour Biol* **40**, 1010428318776822 (2018).
15. Baker, A.M., *et al.* The role of lysyl oxidase in SRC-dependent proliferation and metastasis of colorectal cancer. *J Natl Cancer Inst* **103**, 407-424 (2011).
16. Baker, A.M., Bird, D., Lang, G., Cox, T.R. & Erler, J.T. Lysyl oxidase enzymatic function increases stiffness to drive colorectal cancer progression through FAK. *Oncogene* **32**, 1863-1868 (2013).
17. Chavez, K.J., Garimella, S.V. & Lipkowitz, S. Triple negative breast cancer cell lines: one tool in the search for better treatment of triple negative breast cancer. *Breast Dis* **32**, 35-48 (2010).
18. Holliday, D.L. & Speirs, V. Choosing the right cell line for breast cancer research. *Breast cancer research : BCR* **13**, 215 (2011).
19. Bergin, A.R.T. & Loi, S. Triple-negative breast cancer: recent treatment advances. *F1000Res* **8**(2019).
20. Tanaka, T., Yamaguchi, J., Shoji, K. & Nangaku, M. Anthracycline inhibits recruitment of hypoxia-inducible transcription factors and suppresses tumor cell migration and cardiac angiogenic response in the host. *J Biol Chem* **287**, 34866-34882 (2012).
21. Brenner, K.A., Corbett, S.A. & Schwarzbauer, J.E. Regulation of fibronectin matrix assembly by activated Ras in transformed cells. *Oncogene* **19**, 3156-3163 (2000).
22. Mana, G., *et al.* PPFIA1 drives active alpha5beta1 integrin recycling and controls fibronectin fibrillogenesis and vascular morphogenesis. *Nature communications* **7**, 13546 (2016).
23. Varadaraj, A., Magdaleno, C. & Mythreye, K. Deoxycholate Fractionation of Fibronectin (FN) and Biotinylation Assay to Measure Recycled FN Fibrils in Epithelial Cells. *Bio Protoc* **8**(2018).
24. Wierzbicka-Patynowski, I., Mao, Y. & Schwarzbauer, J.E. Analysis of fibronectin matrix assembly. *Curr Protoc Cell Biol* **Chapter 10**, Unit 10 12 (2004).
25. Schutze, F., *et al.* Inhibition of Lysyl Oxidases Improves Drug Diffusion and Increases Efficacy of Cytotoxic Treatment in 3D Tumor Models. *Scientific reports* **5**, 17576 (2015).

26. Boufraqueh, M., *et al.* Lysyl Oxidase (LOX) Transcriptionally Regulates SNAI2 Expression and TIMP4 Secretion in Human Cancers. *Clinical cancer research : an official journal of the American Association for Cancer Research* **22**, 4491-4504 (2016).

REVIEWERS' COMMENTS:

Reviewer #1 (Remarks to the Author):

The authors show that the majority of the reviewers' comments were adequately addressed in the revised manuscript. However, the authors should address and/or remove reference 29 in their revised manuscript due to the recent retraction of Erler JT et al. Nature (2006).

Retraction Note: Lysyl oxidase is essential for hypoxia-induced metastasis.

Erler JT, Bennewith KL, Nicolau M, Dornhöfer N, Kong C, Le QT, Chi JA, Jeffrey SS, Giaccia AJ. Nature. 2020 Mar;579(7799):456. doi: 10.1038/s41586-020-2112-4.

PMID: 32188947 [PubMed - in process]

Reviewer #2 (Remarks to the Author):

Minor point:

In the authors rebuttal to our comment #6: "We also analyzed additional patient datasets to further confirm the effect of miR-142-3p on the survival of chemotherapy-treated TNBC patients. These datasets include GSE19536, GSE22220, GSE75685 and GSE40267. However, due to the low number of chemotherapy-treated TNBC patients with miRNA expression profiling (i.e., around 30-40 patients in each), unlike METABRIC dataset from which we were able to stratify 106 chemotherapy-treated TNBC patients with miRNA expressions, we did not get a significant result."

The reviewer understands the lack sufficient samples regarding miRNA profiling and associated survival outcomes in chemotherapy-treated TNBC datasets. Although the results were negative and underpowered, the authors should include this in their discussion that miRNA profiling potentially deserves further study and should be investigated more widely in chemotherapy-treated breast cancer cohorts to better understand the clinical impact of miRNA and treatment response for TNBC.

Reviewer #3 (Remarks to the Author):

In their reviewed manuscript the authors have addressed most of my previous concerns. They have added new data to strengthen their results, such as the results generated using other breast cancer cell lines which allowed them to conclude that their proposed mechanism is associated with TNBC cells. However, LOX expression differences between TNBC and non-TNBC are minor, thus, it is an overstatement to say that LOX is expressed at high levels in TNBC.

They also back their hypothesis with results obtained using PDXs and organoids as well as several additional analyses in TNBC datasets and samples.

Although the manuscript has clearly improved after the revision there are still several concerns regarding my previous comments. These concerns are raised based on two major facts: 1) the authors rely most of their results on the use of an inhibitor (BAPN) that is not specific for LOX activity; and 2) they perform LOX silencing assays that are not achieving strong LOX silencing. LOXL2 has been also shown to impact on FAK signaling and it is upregulated under hypoxia. Besides, it has been associated with chemoresistance as well (ref 30 cited by authors refers to LOXL2 specifically).

Regarding new data, the authors claim that LOX activity is upregulated in response to hypoxia due to increased LOX expression. Thus, the sentence in line 209 is not correct: "increased expression of LOX led to its elevated enzymatic activity" since this assay does not discriminate among enzymes. The results shown in Fig. 2k show increased lysyl oxidase activity that could be due to other LOX family members since the authors have not checked the upregulation of other LOX enzymes in hypoxia. Why BAPN was not added to hypoxic conditions? The authors should rephrase

and provide an explanation for the latter in order to confirm their hypothesis that LOX and only LOX is activated under hypoxia promoting integrin signaling. Besides, they show that the expression of other LOX family members is not altered in their doxorubicin-resistant xenografts but the authors never show any data concerning LOX members' expression in TNBC patients' datasets to discard their association with resistance.

Moreover: the authors have checked the expression of LOXL2 in 4T1 cells and it is similar to LOX (answer to q. 12 reviewer #3), suggesting that BAPN effects might be due to LOXL2 instead of LOX inhibition. What about the PDX model they chose to study chemoresistance and LOX (Fig. 5)? Is BAPN inhibiting only LOX? These points need to be addressed to support the conclusions.

In summary, I believe the results do not undoubtedly demonstrate that LOX is indeed responsible for the chemoresistance in TNBC patients, leaving room for concern about the contribution of other LOX family members.

Other points.

line 395: by luciferase assays direct binding is not demonstrated.

It will be important to show protein levels (immunohistochemistry or Western Blot), and not only mRNA between sensitive and resistant tumors in Supp Fig S3A.

ANSWERS TO REVIEWERS' COMMENTS

Reviewer #1 (Remarks to the Author):

The authors show that the majority of the reviewers' comments were adequately addressed in the revised manuscript. However, the authors should address and/or remove reference 29 in their revised manuscript due to the recent retraction of Erler JT et al. Nature (2006).

Retraction Note: Lysyl oxidase is essential for hypoxia-induced metastasis. Erler JT, Bennewith KL, Nicolau M, Dornhöfer N, Kong C, Le QT, Chi JA, Jeffrey SS, Giaccia AJ. Nature. 2020 Mar;579(7799):456. doi: 10.1038/s41586-020-2112-4. PMID: 32188947 [PubMed – in process]

Response: We would like to thank the reviewer once again for evaluating our work. As suggested by the reviewer, we removed this reference from the manuscript, and instead we added another paper showing the role of LOX in metastasis (Rachman-Tzemah, C., *et al. Cell reports* **19**, 774-784 (2017)). This now provided in Page 5, Line 107 and Page 17, Line 407 in the revised manuscript.

Reviewer #2 (Remarks to the Author):

Minor point:

In the authors rebuttal to our comment #6: “We also analyzed additional patient datasets to further confirm the effect of miR-142-3p on the survival of chemotherapy-treated TNBC patients. These datasets include GSE19536, GSE22220, GSE75685 and GSE40267. However, due to the low number of chemotherapy-treated TNBC patients with miRNA expression profiling (i.e., around 30-40 patients in each), unlike METABRIC dataset from which we were able to stratify 106 chemotherapy-treated TNBC patients with miRNA expressions, we did not get a significant result.”

The reviewer understands the lack sufficient samples regarding miRNA profiling and associated survival outcomes in chemotherapy-treated TNBC datasets. Although the results were negative and underpowered, the authors should include this in their discussion that miRNA profiling potentially deserves further study and should be investigated more widely in chemotherapy-treated breast cancer cohorts to better understand the clinical impact of miRNA and treatment response for TNBC.

Response: We would like to thank the reviewer once again for evaluating our work. As suggested by the reviewer, we now discussed that in order to fully understand the contribution of miR-142-3p to chemoresistance in TNBCs, additional analyses with large chemotherapy-treated patient cohorts are required. In this line, miRNA expression profiling or in-situ hybridization in tumors will shed light on the endogenous expression pattern of miR-142-3p in tumors and uncover the potential of this miRNA as a biomarker and a therapy sensitizer in TNBCs. This now provided in Page 20, Lines 467-471 in the revised manuscript.

Reviewer #3 (Remarks to the Author):

In their reviewed manuscript the authors have addressed most of my previous concerns. They have added new data to strengthen their results, such as the results generated using other breast cancer cell lines which allowed them to conclude that their proposed mechanism is associated with TNBC cells. However, LOX expression differences between TNBC and non-TNBC are minor, thus, it is an overstatement to say that LOX is expressed at high levels in TNBC. They also back their hypothesis with results obtained using PDXs and organoids as well as several additional analyses in TNBC datasets and samples.

Response: We would like to thank the reviewer for appreciating the efforts we made to strengthen the manuscript by implementing the constructive requests from the reviewers, including PDXs and organoids as well as additional analysis in TNBC datasets and samples. During the first revision, we analyzed LOX expression among different subtypes of breast cancer cell lines and stated that “TNBC cells express higher levels of LOX” based on the western blot analysis. We have now softened this statement as requested by the reviewer by saying that “TNBC cells express relatively higher levels of LOX”. This now provided in Page 11, Lines 243 in the revised manuscript.

Although the manuscript has clearly improved after the revision there are still several concerns regarding my previous comments. These concerns are raised based on two major facts: 1) the authors rely most of their results on the use of an inhibitor (BAPN) that is not specific for LOX activity; and 2) they perform LOX silencing assays that are not achieving strong LOX silencing. LOXL2 has been also shown to impact on FAK signaling and it is upregulated under hypoxia. Besides, it has been associated with chemoresistance as well (ref 30 cited by authors refers to LOXL2 specifically).

Response: We would like to thank the reviewer for finding our manuscript significantly improved after the revision. Regarding the concerns of the reviewer on the non-specific nature of BAPN as a LOX family inhibitor, we toned down most of our sentences by stating that BAPN is not a LOX specific inhibitor, but rather a LOX family inhibitor (Page 9, Line 208; Page 11, Line 244; Page 19, Lines 450-451). Regarding LOX silencing, in the initial version, we had provided data from two different siRNAs that confer different levels of knockdown and we showed a dose-dependent decrease in integrin signaling activity. During the first revision, we provided additional data using shLOX expressing cells where we demonstrated a stronger silencing than siRNAs. This was achieved using two different shRNAs and importantly, we have seen decreased ITGA5 expression and reduced activity of downstream signaling upon LOX knockdown with these shRNAs (Figure 2n). These *in vitro* findings have also been recapitulated *in vivo* where we were able to potentiate doxorubicin response upon LOX knockdown which was accompanied by decreased ITGA5 expression and downstream activity (Figure 6). Importantly, we observed no decrease in the levels of LOXL2 or the other LOX family members in shLOX-expressing tumors (Supplementary Figure 11e, f), strongly suggesting that the increase in doxorubicin response is specifically due to the decrease in LOX levels in these models. In addition to these loss-of-function experiments, we also demonstrated that overexpressing the LOX enzyme, specifically, confers doxorubicin resistance in collagen-embedded cells and also led to increased ITGA5 expression (Figure 3d, e). Overall, these data together with the specific overexpression of only the LOX enzyme in our chemoresistant TNBC model suggest that LOX expression leads to chemoresistance in TNBCs. However, in agreement with the reviewer, we still do not underestimate the potential effects of LOXL2 inhibition on chemotherapy response in different cancers as it has been previously demonstrated to be a sensitizer to gemcitabine-based chemotherapy in pancreatic ductal adenocarcinoma (Le Calvé B. et al, 2016) as we cited in our manuscript.

Regarding new data, the authors claim that LOX activity is upregulated in response to hypoxia due to increased LOX expression. Thus, the sentence in line 209 is not correct: “increased expression of LOX led to its elevated enzymatic activity” since this assay does not discriminate among enzymes. The results shown in Fig. 2k show increased lysyl oxidase activity that could be due to other LOX family members since the authors have not checked the upregulation of other LOX enzymes in hypoxia. Why BAPN was not added to hypoxic conditions? The authors should rephrase and provide an explanation for the latter in order to confirm their hypothesis that LOX and only LOX is activated under hypoxia promoting integrin signaling. Besides, they show that the expression of other LOX family members is not altered in their doxorubicin-resistant xenografts but the authors never show any data concerning LOX members’ expression in TNBC patients’ datasets to discard their association with resistance.

Response: Regarding the concerns of the reviewer on the LOX activity assay we performed in normoxia versus hypoxia, we agree with the reviewer that LOX activity assay is not able to discriminate among different LOX enzymes. Accordingly, we rephrased our conclusion on this assay, and we now state that “LOX enzymatic activity was higher under hypoxia as compared to normoxia, potentially due to increased LOX expression” (Page 9, Line 207). We believe that this statement now adequately summarizes our results on increased mRNA and protein levels of LOX and the increased lysyl oxidase activity under hypoxia. We also revised the next sentence by saying that “BAPN, a LOX family inhibitor (instead of a LOX inhibitor) is used as a negative control” (Page 9, Line 208). However, these data should not be interpreted in isolation from the siRNA or shRNA-mediated knockdowns of LOX and their effects on the downstream signaling which was induced by hypoxia.

Regarding the concerns on the potential contribution of LOXL2, which is a LOX family member that has previously been studied in the context of chemoresistance, we further examined the expression of LOXL2 among chemotherapy-treated TNBC patients with pathologic complete response (pCR) versus residual disease (RD) from GSE25066 for the second revision. This is the dataset where we showed that LOX, ITGA5 and FN1 are expressed at higher levels in patients with RD as compared to patients with pCR (Supplementary Fig. 5). Among the same patients, we observed no significant association of LOXL2 expression with chemotherapy response (Figure for Reviewer). This data is also in line with our RNA-sequencing analyses where we observed a significant increase in LOX expression and no change in LOXL2 levels in our doxorubicin-resistant xenografts (Supplementary Fig. 3).

Figure for Reviewer. mRNA expression of LOXL2 in TNBC patients with pCR vs. RD from GSE25066.

Moreover: the authors have checked the expression of LOXL2 in 4T1 cells and it is similar to LOX (answer to q. 12 reviewer #3), suggesting that BAPN effects might be due to LOXL2 instead of LOX inhibition. What about the PDX model they chose to study chemoresistance and LOX (Fig. 5)? Is BAPN inhibiting only LOX? These points need to be addressed to support the conclusions. In summary, I believe the results do not undoubtedly demonstrate that LOX is indeed responsible for the chemoresistance in TNBC patients, leaving room for concern about the contribution of other LOX family members.

Response: Although we observed no association of LOXL2 expression with TNBC chemoresistance, we still agree with the reviewer that in assays where we used BAPN, a LOX family inhibitor, inhibition of LOXL2 might also be contributing to the overall effect to a certain extent, since LOXL2 is expressed at similar levels with LOX in 4T1 model. This is also true for the PDX model which expresses similar levels of LOX and LOXL2. However, it should also be noted that hypoxia and focal adhesion signaling activity positively correlates with LOX expression in our list of chemotherapy-treated TNBC PDXs, and that our selection was based on a criterion of expressing high levels of LOX and therefore higher levels of hypoxia and focal adhesion signaling. During the revision, we showed the presence of LOX in each of these models with either western blot or IHC analysis to make sure that BAPN inhibits LOX activity. As discussed above, we now make emphasis on the non-specific nature of BAPN, the most widely used LOX family inhibitor, and revised the sentences in the manuscript where we mentioned BAPN as a LOX

inhibitor accordingly (Page 9, Line 208; Page 11, Line 243; Page 19, Lines 449-450). Based on the ample amount of literature on different LOX family members in different disease contexts, especially the LOX and LOXL2 enzymes, we underline the importance of designing inhibitors that specifically target different LOX family members, i.e. LOX and LOXL2, so that we take advantage of inhibiting these enzymes in models where each of them plays important roles depending on the disease context (Page 19, Lines 449-453). This is exactly what we are currently working on, i.e., designing LOX-specific inhibitors to overcome chemoresistance in TNBC patients in the future.

Other points.

Line 395: by luciferase assays direct binding is not demonstrated.

Response: The dual luciferase assay is a very well-established state-of-art method to detect the direct binding between ectopically expressed miRNAs and the 3'-UTR regions of their predicted targets^{1, 2}. This method has been used in many different studies by others^{3, 4} and us^{5, 6, 7, 8} as a measure of direct binding between miRNA-mRNA pairs. The assay is based on cloning the 3'-UTR region of the predicted mRNA target into a vector (e.g. psiCHECK-2 vector) that contains a *Renilla* luciferase gene as the reporter enzyme. This vector also has *Firefly* luciferase gene allowing normalization among different groups. Cells are co-transfected with this vector and the miRNA mimic vs. the non-targeting miRNA (miR-Negative) as the control, and dual luciferase activity is measured. Direct binding of the miRNA to the 3'-UTR region leads to reduced luciferase reporter protein, resulting in a decrease in relative *Renilla/Firefly* luciferase activity as compared to miR-Negative-transfected condition.

It will be important to show protein levels (immunohistochemistry or Western Blot), and not only mRNA between sensitive and resistant tumors in Supp Fig S3A.

Response: We had already provided those data in Figures 2e, 1g and 1j in our manuscript.

References:

1. Clement T, Salone V, Rederstorff M. Dual luciferase gene reporter assays to study miRNA function. *Methods in molecular biology* **1296**, 187-198 (2015).
2. Jin Y, Chen Z, Liu X, Zhou X. Evaluating the microRNA targeting sites by luciferase reporter gene assay. *Methods in molecular biology* **936**, 117-127 (2013).
3. Bracken CP, *et al.* A double-negative feedback loop between ZEB1-SIP1 and the microRNA-200 family regulates epithelial-mesenchymal transition. *Cancer research* **68**, 7846-7854 (2008).
4. Gregory PA, *et al.* The miR-200 family and miR-205 regulate epithelial to mesenchymal transition by targeting ZEB1 and SIP1. *Nature cell biology* **10**, 593-601 (2008).
5. Jurmeister S, *et al.* MicroRNA-200c represses migration and invasion of breast cancer cells by targeting actin-regulatory proteins FHOD1 and PPM1F. *Molecular and cellular biology* **32**, 633-651 (2012).
6. Uhlmann S, *et al.* Global microRNA level regulation of EGFR-driven cell-cycle protein network in breast cancer. *Molecular systems biology* **8**, 570 (2012).
7. Ward A, *et al.* Re-expression of microRNA-375 reverses both tamoxifen resistance and accompanying EMT-like properties in breast cancer. *Oncogene* **32**, 1173-1182 (2013).
8. Ward A, *et al.* MicroRNA-519a is a novel oncomir conferring tamoxifen resistance by targeting a network of tumour-suppressor genes in ER+ breast cancer. *The Journal of pathology* **233**, 368-379 (2014).